# FANCJ helicase promotes DNA end resection by facilitating CtIP recruitment to DNA double-strand breaks

**Sarmi Nath**, **Ganesh Nagaraju** *

Department of Biochemistry, Indian Institute of Science, Bangalore, India

* nganesh@iisc.ac.in

## Abstract

FANCJ helicase mutations are known to cause hereditary breast and ovarian cancers as well as bone marrow failure syndrome Fanconi anemia. FANCJ plays an important role in the repair of DNA inter-strand crosslinks and DNA double-strand breaks (DSBs) by homologous recombination (HR). Nonetheless, the molecular mechanism by which FANCJ controls HR mediated DSB repair is obscure. Here, we show that FANCJ promotes DNA end resection by recruiting CtIP to the sites of DSBs. This recruitment of CtIP is dependent on FANCJ K1249 acetylation. Notably, FANCJ acetylation is dependent on FANCJ S990 phosphorylation by CDK. The CDK mediated phosphorylation of FANCJ independently facilitates its interaction with BRCA1 at damaged DNA sites and promotes DNA end resection by CtIP recruitment. Strikingly, mutational studies reveal that ATP binding competent but hydrolysis deficient FANCJ partially supports end resection, indicating that in addition to the scaffolding role of FANCJ in CtIP recruitment, its helicase activity is important for promoting end resection. Together, these data unravel a novel function of FANCJ helicase in DNA end resection and provide mechanistic insights into its role in repairing DSBs by HR and in genome maintenance.

## Author summary

Homologous recombination has been considered as an error-free pathway in repairing DSBs and maintaining genome stability. Cyclin-dependent kinases (CDKs) and various factors including MRE11, CtIP, EXO1, and BLM helicase participate in DNA end resection to promote HR mediated DSB repair. Despite the identification of FANCJ helicase role in HR and tumor suppression, the molecular mechanism by which FANCJ helicase participates in HR is obscure. Here, we show that FANCJ helicase controls DNA end resection by recruiting CtIP to the sites of DSBs. The loading of CtIP is dependent on FANCJ acetylation which is mediated by CDK dependent phosphorylation of FANCJ. Moreover, in addition to FANCJ mediated CtIP recruitment, its helicase activity is also essential for DNA end resection. Our data identify FANCJ as a novel player in the DNA end resection and provide insights into its role in HR mediated DSB repair.

**Data Availability Statement:** All relevant data are within the manuscript and its Supporting Information files.

**Funding:** Funding by Department of Science and Technology (EMR/2015/001720); Department of

Biotechnology (DBT) (BT/PR23498/BRB/10/1590/
2017), DBT-National Bioscience Award (BT/HRD/
NBA/37/01/2015), IISc-DBT partnership program
(DBT/BF/PR/INS/IISc/2011–12 and BT/PR27952/
INF/22/212/2018) and infrastructure support
provided by funding from DST and UGC are greatly
acknowledged. S.N. is supported by fellowship
from IISc and DBT. The funders had no role in
study design, data collection and analysis, decision
to publish, or preparation of the manuscript.

**Competing interests:** The authors have declared
that no competing interests exist.

## Introduction

DNA double-strand breaks (DSBs) are the most toxic form of DNA lesions that arise endoge-
nously or by exposure to various chemicals and radiation. Unrepaired or mis-repaired DSBs
can lead to genome instability and tumorigenesis [1]. Indeed, defect in the repair of DSBs is
associated with chromosome instability and cancer susceptibility genetic diseases including
Fanconi anemia (FA), Bloom syndrome and Werner syndrome [2, 3]. There are two major
pathways to repair DSBs: non-homologous end joining (NHEJ) and homologous recombina-
tion (HR). NHEJ mediated repair is DNA template-independent and occurs in all phases of
the cell cycle, and is often error-prone. In contrast, HR mediated DSB repair is mostly error-
free, requires homologous template DNA and is restricted to S and G2 phase. In somatic cells,
sister chromatids serve as a template for repairing broken DNA; thus, HR is considered as the
most accurate pathway for repairing DSBs and maintaining genome integrity [4–8].

Repair of DSBs by HR requires resection of DNA ends which are generated by nucleases
and helicases [9]. Resection of DNA ends occurs in two steps; in the first step, MRE11 by its
endonuclease and exonuclease activities generates short 3′-single stranded DNA (ssDNA)
overhangs. RAD50, NBS1, and CtIP stimulate MRE11 mediated short-range resection. In the
second step, long-range resection is catalysed by either EXO1 or DNA2 nucleases in conjunc-
tion with BLM or WRN helicase [10–12]. The 3′ ssDNA overhangs are coated by RPA protein
which is subsequently replaced by RAD51 recombinase with the assistance of mediator pro-
teins including BRCA1, BRCA2 and RAD51 paralogs [3, 13, 14]. RPA coated ssDNA also
recruits ATRIP to execute ATR mediated checkpoint activation [15]. Thus, resection of DNA
ends is crucial for determining DNA repair pathway choice and checkpoint activation.

FANCJ helicase mutations lead to bone marrow failure syndrome Fanconi anemia (FA),
and breast and ovarian cancers [16, 17]. Evidence from various studies indicate that FANCJ is
a multifunctional helicase that participates in the FA pathway of DNA inter-strand crosslink
(ICL) repair [17], DSB repair by HR [18, 19], G-quadruplex DNA resolution [20], in the rescue
of cells from UV induced lesions [21] and in the maintenance of microsatellite stability [22].
FANCJ also has been shown to protect replication forks during replication stress [23]. Our
previous study showed that FANCJ plays an important role in regulating the balance between
short and long-tract gene conversions in response to I-*Sce*I induced breaks and this function
appears to be independent of its role in ICL repair [19]. However, the precise mechanism by
which FANCJ regulates HR is obscure. Here, we show that FANCJ promotes DNA end resec-
tion by recruiting CtIP to the sites of DSBs, which is dependent on FANCJ K1249 acetylation.
Strikingly, the acetylation of FANCJ is dependent on FANCJ S990 phosphorylation by CDK.
FANCJ interacts with BRCA1 upon phosphorylation by CDK and promotes DNA end resec-
tion in a manner independent of the BRCA1-CtIP complex. Additionally, FANCJ-CtIP medi-
ated end resection requires FANCJ helicase activity. Together, our data identify a novel
function of FANCJ helicase in DNA end resection to promote HR mediated DSB repair.

## Results

### FANCJ helicase is required for processing DSB ends

To gain insights into the role of FANCJ in HR, we measured ssDNA generation at sites of
DSBs induced by *Asi*SI restriction enzyme by a previously developed ER-*Asi*SI system in
U2OS cells [24]. Incubation of cells with 4-OHT facilitates the entry of *Asi*SI enzyme into
nucleus and DSB generation at multiple sites in the genome. This system allows quantitative
measurement of ssDNA generation at *Asi*SI induced DSB1 and DSB2 sites in chromosome 1
(S1A Fig). To test the role of FANCJ in DNA end resection, we incubated U2OS cells with

4-OHT and measured ssDNA generation qualitatively by measuring BrdU staining in CENP-F positive cells which is specific to S/G2 phase (S1B Fig). Compared to control cells, depletion of end resection factors CtIP, MRE11 and BLM caused a defect in the BrdU signal (S1C Fig and Fig 1A and 1B). Interestingly, the depletion of FANCJ caused ~3-fold reduction in BrdU intensity. RPA2 phosphorylation at serine 4 and serine 8 residues serves as a marker for ssDNA generation in the genome [25]. Compared to control cells, FANCJ deficient cells exhibited ~3-fold reduction in phosphorylated RPA2 in response to *Asi*SI and zeocin induced breaks, indicating the possible role of FANCJ in DNA end resection (Fig 1C and 1D, S2A and S2B Fig). This was further confirmed by immunoblotting for phosphorylated RPA2 in control and FANCJ/CtIP depleted cells (Fig 1E and S1C Fig). Processing of DSB ends facilitates the loading of RAD51 onto the ssDNA to initiate HR [13]. To test whether FANCJ deficiency affects RAD51/RPA loading onto the sites of DSBs, we measured RAD51 and RPA70 foci formation after inducing breaks with *Asi*SI. Compared to control cells, FANCJ/CtIP depleted cells exhibited reduced RAD51 and RPA70 foci formation (Fig 1F–1I). Similar to *Asi*SI induced breaks, localization/loading of RPA70 and RAD51 was impaired in FANCJ depleted U2OS cells upon generation of DSBs induced by etoposide and zeocin (S2D–S2I Fig). Collectively, these data suggest that FANCJ participates in DNA end resection.

To validate our observation of FANCJ role in DNA end resection, we quantitatively measured ssDNA generation at *Asi*SI induced DSB1 and DSB2 sites [24]. We carried out quantitative PCR using a set of primers that measure ssDNA generation ranging from ~130bp to ~3.5kb (S1A Fig and S1 Table). Compared to control cells, depletion of MRE11, CtIP, and BRCA1 caused 2–3 fold reduction in ssDNA generation at DSB1 and DSB2 sites (Fig 1J and S1C Fig). These results are in agreement with previous observations of defective end resection associated with factors that participate in DSB processing [11, 12]. Consistent with DNA2 helicase/nuclease role in the long-range resection [26], we observed moderate reduction in ssDNA generation close to the DSB sites but showed a significant reduction at the range of 1.6–3.5 kb (Fig 1J). 53BP1 is an upstream regulator in the pathway choice of DSB repair and its depletion is known to cause an increase in the end resection [27, 28]. In agreement with this, we find an increase in DNA end resection upon depletion of 53BP1 (Fig 1J). Strikingly, the depletion of FANCJ helicase by two shRNA plasmids caused ~2-fold defect in ssDNA generation at both DSB1 and DSB2 sites (Fig 1J). Together, these results suggest that FANCJ plays an important role in DNA end resection.

## FANCJ interacts with CtIP and facilitates its recruitment

MRN complex in collaboration with CtIP initiates the processing of DSB ends to generate short tracts of ssDNA [29, 30]. Subsequently, EXO1 and DNA2 nucleases in conjunction with BLM/WRN helicase promote long-range resection [26, 31, 32]. To gain insights into the role of FANCJ in DNA end resection, we investigated the recruitment of FANCJ to the DSB1 site and the role of end resection components in the localization of FANCJ. Compared to control cells, induction of DSBs with *Asi*SI resulted in ~4-fold enrichment of FANCJ to the sites of DSBs and this enrichment was moderately affected in MRE11 depleted cells (Fig 2A). However, BRCA1 depletion caused a more significant reduction in FANCJ localization but depletion of CtIP or BLM did not affect FANCJ recruitment (Fig 2A). To validate our observations, we measured FANCJ foci formation in response to *Asi*SI and zeocin induced DSBs in control, BRCA1 and MRE11 depleted cells. Compared to control cells, MRE11/BRCA1 depleted cells exhibited up to an 8-fold reduction in FANCJ localization at the sites of DSBs (Fig 2B and 2C, S2J and S2K Fig).

The localization of FANCJ to the sites of DSBs prompted us to investigate whether FANCJ interacts with end resection components before and after inducing DSBs by *Asi*SI. Nuclear

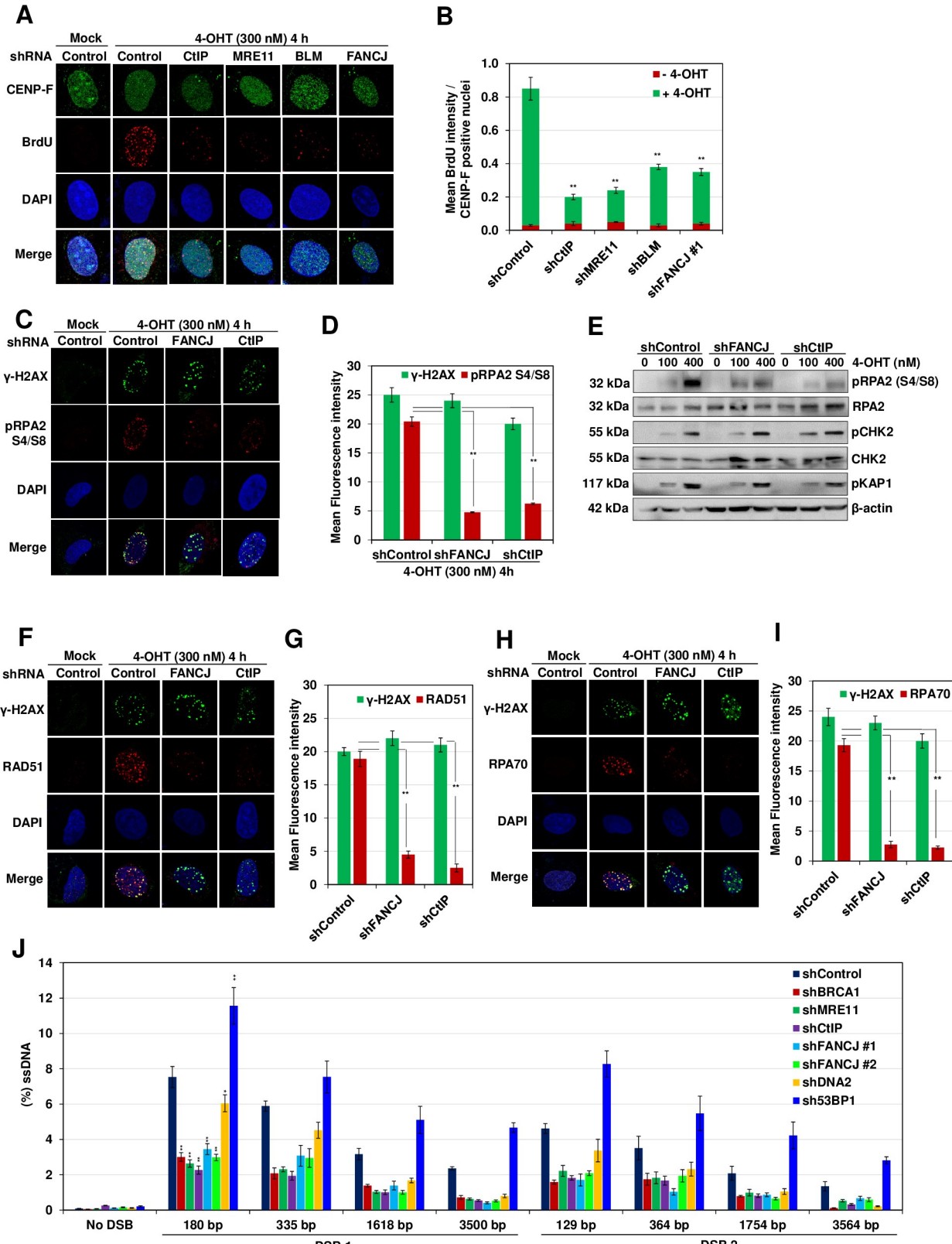

**Fig 1. FANCJ facilitates DNA end resection.** (A) To measure end resection by BrdU staining, ER-*Asi*SI U2OS cells pre-labelled with BrdU for 24 h were depleted for the indicated proteins followed by treatment with 300 nM 4-OHT for 4 h or mock treated. The efficiency of knockdown of

individual proteins was examined by western blotting (S1C Fig). Cells were fixed and stained with BrdU antibodies in native conditions to selectively detect ssDNA generated by end resection. CENP-F was used as an S/G2 phase marker to specifically examine DNA end resection during HR. Representative image for BrdU foci is shown. (B) Graph represents mean BrdU intensity from indicated samples in (A). N = 3; error bars indicate standard deviation (SD) and statistical significance was measured by two-tailed *Student*'s t-test of unequal variance. *p < 0.05; **p < 0.01; ***p < 0.001; N.S., non-significant. (C) ER-*Asi*SI U2OS cells depleted of the indicated proteins were treated with 300 nM 4-OHT for 4 h or mock treated. Cells were fixed and stained with γ-H2AX and pRPA2 (S4/S8) antibodies to detect ssDNA generated by end resection. Representative image for γ-H2AX and pRPA2 (S4/S8) foci are shown. (D) Graph represents the mean fluorescence intensity of γ-H2AX and pRPA2 (S4/S8) foci/nucleus from indicated cells in (C). N = 3; error bars indicate standard deviation (SD) and statistical significance was measured by two-tailed *Student*'s t-test of unequal variance. *p < 0.05; **p < 0.01; ***p < 0.001; N.S., non-significant. (E) ER-*Asi*SI U2OS cells treated with either control shRNA, shFANCJ #1 or shCtIP were treated with increasing dose of 4-OHT (0,100 and 400 nM) for 4 h. Whole cell lysates were resolved on 10% SDS-PAGE and probed for the indicated proteins to measure their damage induced enrichment in the cell. (F) ER-*Asi*SI U2OS cells depleted for the indicated proteins were treated with 300 nM 4-OHT for 4 h or mock treated. Cells were fixed and stained with γ-H2AX and RAD51 antibodies. Representative image for γ-H2AX and RAD51 foci are shown. (G) Graph represents the mean fluorescence intensity of γ-H2AX and RAD51 foci/nucleus from indicated cells in (F). N = 3; error bars indicate standard deviation (SD) and statistical significance was measured by two-tailed *Student*'s t-test of unequal variance. *p < 0.05; **p < 0.01; ***p < 0.001; N.S., non-significant. (H) ER-*Asi*SI U2OS cells depleted for the indicated proteins were treated with 300 nM 4-OHT for 4 h or mock treated. Cells were fixed and stained with γ-H2AX and RPA70 antibodies. Representative image for γ-H2AX and RPA70 foci are shown. (I) Graph represents the mean fluorescence intensity of γ-H2AX and RPA70 foci/nucleus from indicated cells in (H). N = 3; error bars indicate standard deviation (SD) and statistical significance was measured by two-tailed *Student*'s t-test of unequal variance. *p < 0.05; **p < 0.01; ***p < 0.001; N.S., non-significant. (J) Measurement of DSB end resection in ER-*Asi*SI U2OS cells transfected with control shRNA or shRNAs directed against BRCA1, FANCJ, CtIP, MRE11, DNA2 and 53BP1 as indicated using the assay established in S1A Fig. The efficiency of knockdown of individual proteins was examined by western blotting (S1C Fig). ER-*Asi*SI U2OS cells depleted of indicated proteins were synchronized in S/G2 phase as depicted in S1B Fig followed by treatment with 300 nM 4-OHT for 4 h or mock treated, genomic DNA (gDNA) was extracted and digested or mock digested with *Ava*I, *Nme*AIII, *Bsr*GI, *Bam*HI or *Hin*dIII overnight. DNA end resection adjacent to DSB1, DSB2 and No DSB site was measured by qPCR as described in 'Materials and Methods' section. N = 3; with error bars indicating SD and statistical significance was measured by two-tailed *Student*'s t-test of unequal variance. Summary of the % DSBs at the two selected *Asi*SI sites are shown in S2 Table. *p < 0.05; **p < 0.01; ***p < 0.001; N.S., non-significant.

fractions of damaged and undamaged cells were subjected to immunoprecipitation (IP) using FANCJ specific antibody. FANCJ immunoprecipitates were subjected to immunoblotting with antibodies to various end resection factors. As reported earlier [33], the interaction of FANCJ with BRCA1, MRE11, BLM, and MLH1 was evident before and after inducing breaks [33–36] (Fig 2D). Notably, FANCJ interacted with CtIP only after inducing DSBs (Fig 2D). Reciprocal IP with CtIP also showed FANCJ association (Fig 2E). These interactions were direct and not mediated by DNA as we performed IP assays in the presence of benzonase.

Next, we examined whether FANCJ affects the recruitment of CtIP and the nucleases MRE11, DNA2 and EXO1, and BLM helicase. Induction of DSBs with *Asi*SI resulted in 4–6 fold enrichment of MRE11 and CtIP to the DSB1 site. Although the depletion of FANCJ did not affect MRE11 localization, FANCJ deficiency caused ~3-fold reduction in CtIP recruitment (Fig 2F). To understand whether FANCJ and CtIP participate in a common pathway of end resection, we measured ssDNA generation by qPCR using a set of primers specific to DSB1 and DSB2 site. The depletion of either FANCJ or CtIP caused a 2–3 fold defect in end resection (Fig 2G). Interestingly, the co-depletion of FANCJ with CtIP did not show any further reduction in end resection compared to FANCJ or CtIP depleted cells (Fig 2G), indicating that FANCJ and CtIP participate in a common pathway in processing the DSBs. These results were further corroborated with HR reporter assay which showed that co-depletion of FANCJ and CtIP does not cause any further reduction in GFP positive cells compared to FANCJ or CtIP depleted cells (Fig 2H). Defect in HR channelizes the breaks for repair via NHEJ [33, 37]. Indeed, FANCJ deficient cells exhibit a moderate but significant increase in NHEJ (Fig 2I).

## CtIP interacts at the C-terminal region of FANCJ

FANCJ interacts with BRCA1, BLM, and MRE11, and this interaction is localized to the C-terminal region (residues 881–1249) of FANCJ [16] (Fig 3A). We speculated that CtIP also may be interacting with the C-terminal domain of FANCJ. To test this, we generated shRNA resistant WT-FANCJ and FANCJ C-terminal truncating mutant (CΔ-FANCJ (1–881)), and examined the interaction of FANCJ and CtIP. In agreement with previous studies [33, 35], FANCJ

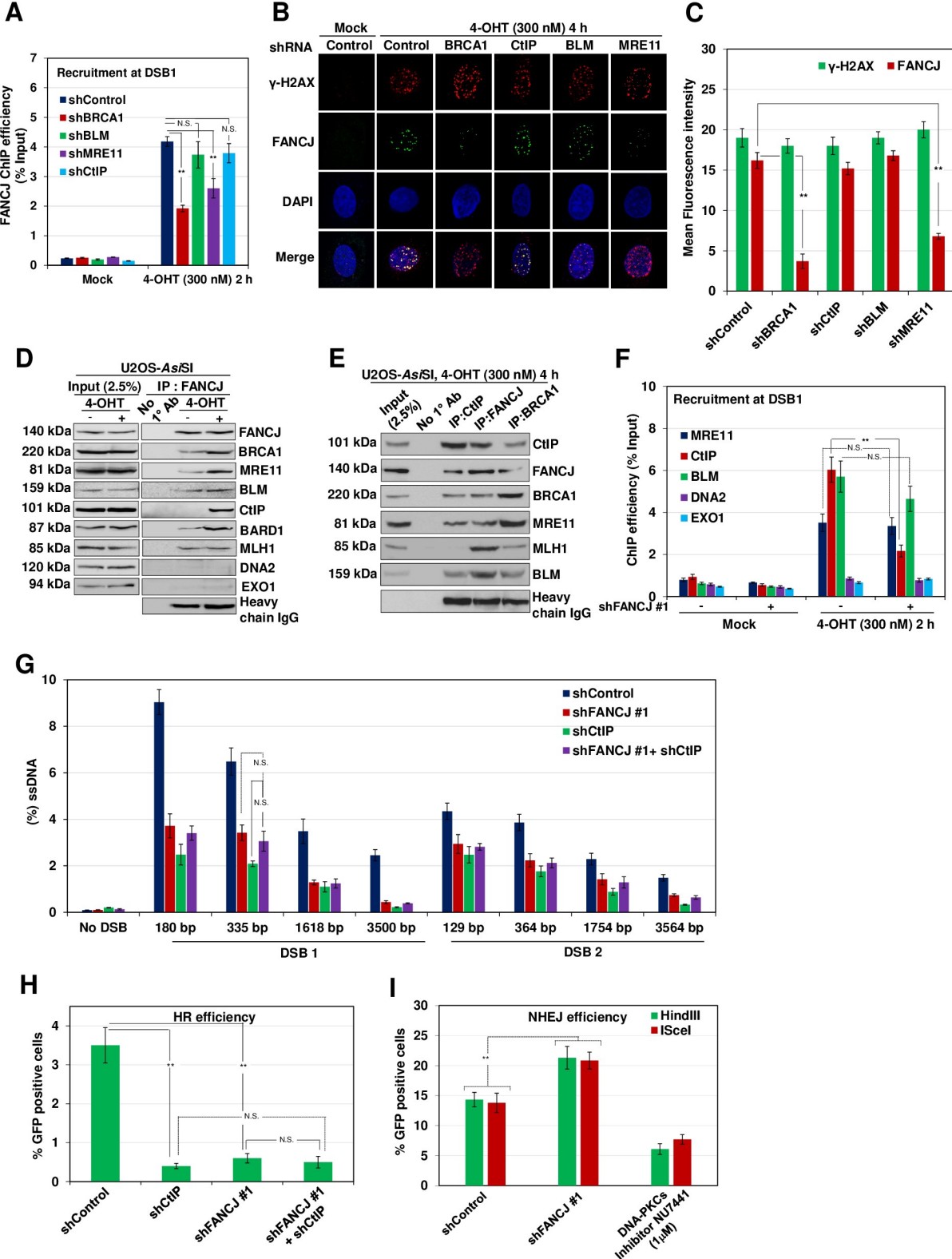

**Fig 2. FANCJ interacts with CtIP in a damage inducible manner and regulates its recruitment to DSBs.** (A) ER-*Asi*SI U2OS cells depleted of the indicated proteins were treated with 300 nM 4-OHT for 2 h or mock treated, and ChIP assay was performed using antibody directed against FANCJ. (B) ER-*Asi*SI U2OS cells depleted for the indicated proteins were treated with 300 nM 4-OHT for 4 h or mock treated. Cells were fixed and stained with γ-H2AX and FANCJ antibodies. Representative image for γ-H2AX and FANCJ foci are shown. (C) Graph represents the mean fluorescence intensity of γ-H2AX and FANCJ foci/nucleus from indicated cells in (B). N = 3; error bars indicate standard

deviation (SD) and statistical significance was measured by two-tailed *Student*'s t-test of unequal variance. $^*p < 0.05$; $^{**}p < 0.01$; $^{***}p < 0.001$; N.S., non-significant. (D) ER-*Asi*SI U2OS nuclear extracts were coimmunoprecipitated with FANCJ antibody before and after treatment with 300 nM 4-OHT for 4 h. The blot was probed with antibodies against the indicated proteins. (E) ER-*Asi*SI U2OS nuclear extracts were coimmunoprecipitated with antibodies against FANCJ, BRCA1 and CtIP after treatment with 300 nM 4-OHT for 4 h. The blot was probed with antibodies against the indicated proteins. Lane 1 represents 2.5% of input and Protein A/G Sepharose beads only were used as negative control. (F) FANCJ depleted ER-*Asi*SI U2OS cells were treated with 300 nM 4-OHT for 2 h or mock treated, and ChIP assays were performed using antibody directed against MRE11, CtIP, DNA2, BLM and EXO1. In both (A) and (F), ChIP efficiencies (as percent of input immunoprecipitated) were measured by semiquantitative PCR at 80 bp from *Asi*SI induced DSB1 site. N = 3, with error bars indicating SD and statistical significance was measured by two-tailed *Student*'s t-test of unequal variance. $^*p < 0.05$; $^{**}p < 0.01$; $^{***}p < 0.001$; N.S., non-significant. (G) ER-*Asi*SI U2OS cells depleted for FANCJ and CtIP, individually and in combination were treated with 300 nM 4-OHT for 4 h or mock treated and measurement of end resection was carried out adjacent to DSB1 and DSB2 sites as previously mentioned. N = 3; with error bars indicating SD and statistical significance was measured by two-tailed *Student*'s t-test of unequal variance. $^*p < 0.05$; $^{**}p < 0.01$; $^{***}p < 0.001$; N.S., non-significant. (H) I-*Sce*I induced GFP+ frequencies (total GFP; overall HR) in U2OS SCR18 cells transfected with shRNAs against FANCJ and CtIP, individually and in combination. $^*p < 0.05$; $^{**}p < 0.01$; $^{***}p < 0.001$; N.S., non-significant. (I) Quantification of NHEJ efficiency in cells treated with DNA PKcs inhibitor NU7441 (1 μM) and control or FANCJ depleted cells transfected with *Hind*III or I-*Sce*I linearized reporter. N = 3; with error bars indicating SD and statistical significance was measured by two-tailed *Student*'s t-test of unequal variance. $^*p < 0.05$; $^{**}p < 0.01$; $^{***}p < 0.001$; N.S., non-significant.

lacking C-terminal 368 residues failed to interact with MRE11, BLM, BRCA1, and BARD1 but not with MLH1 (Fig 3B). Interestingly, deletion of the C-terminal region abolished the interaction of FANCJ with CtIP (Fig 3B), suggesting that C-terminal 368 residues in FANCJ are critical for its interaction with CtIP.

Next, we examined whether FANCJ C-terminal mutant is competent to load on to the sites of DSBs. Consistent with earlier data (Fig 2A), the induction of DSBs with *Asi*SI caused ~10 fold increase in the localization of FANCJ to the DSB1 site (Fig 3C). However, under similar conditions, CΔ-FANCJ failed to localize to the *Asi*SI induced DSB site, suggesting that FANCJ recruitment to the sites of DSBs is mediated via the C-terminal region of FANCJ. Failure in the localization of CΔ-FANCJ also affected ssDNA generation at DSB1 and DSB2 site (Fig 3D). Similarly, ~6 fold reduction in BrdU intensity was evident in cells expressing CΔ-FANCJ compared to WT cells (Fig 3E and 3F). Consistent with end resection defect, FANCJ deficient or CΔ-FANCJ expressing cells exhibited a high degree of cell death compared to control cells in response to *Asi*SI induced breaks (Fig 3G).

## Phosphorylation of FANCJ is essential for CtIP interaction and to promote DNA end resection

FANCJ S990 has been shown to be a phosphorylation target by CDK [38]. To investigate whether FANCJ phosphorylation is important for CtIP association and end resection, we expressed shRNA resistant FANCJ S990A phosphodeficient and FANCJ S990E phosphomimetic mutant and studied the interaction of FANCJ with CtIP (S3A Fig). Compared to control cells, expression of FANCJ S990A mutant abolished the physical association with CtIP and BRCA1 (Fig 4A). Similarly, this mutant was defective in its ability to localize to DSB1 and facilitate end resection (Fig 4B and 4C). Consistently, a significant reduction in BrdU positive cells was evident in cells expressing FANCJ phosphomutant (Fig 4D and 4E). In contrast, FANCJ S990E was competent to bind with CtIP and BRCA1 and promoted efficient ssDNA generation at DSB1 and DSB2 site (Fig 4A–4E). In agreement with these observations, FANCJ phosphomutant cells displayed survival defect compared to control cells in response to *Asi*SI induced DSBs (Fig 4F). These data suggest that FANCJ phosphorylation is crucial for its interaction with CtIP and to promote DNA end resection.

## FANCJ acetylation is crucial for its interaction with CtIP and for end resection

FANCJ K1249 residue has been shown to undergo acetylation and this modification has been implicated in DNA end processing and checkpoint activation [39]. To gain further insights

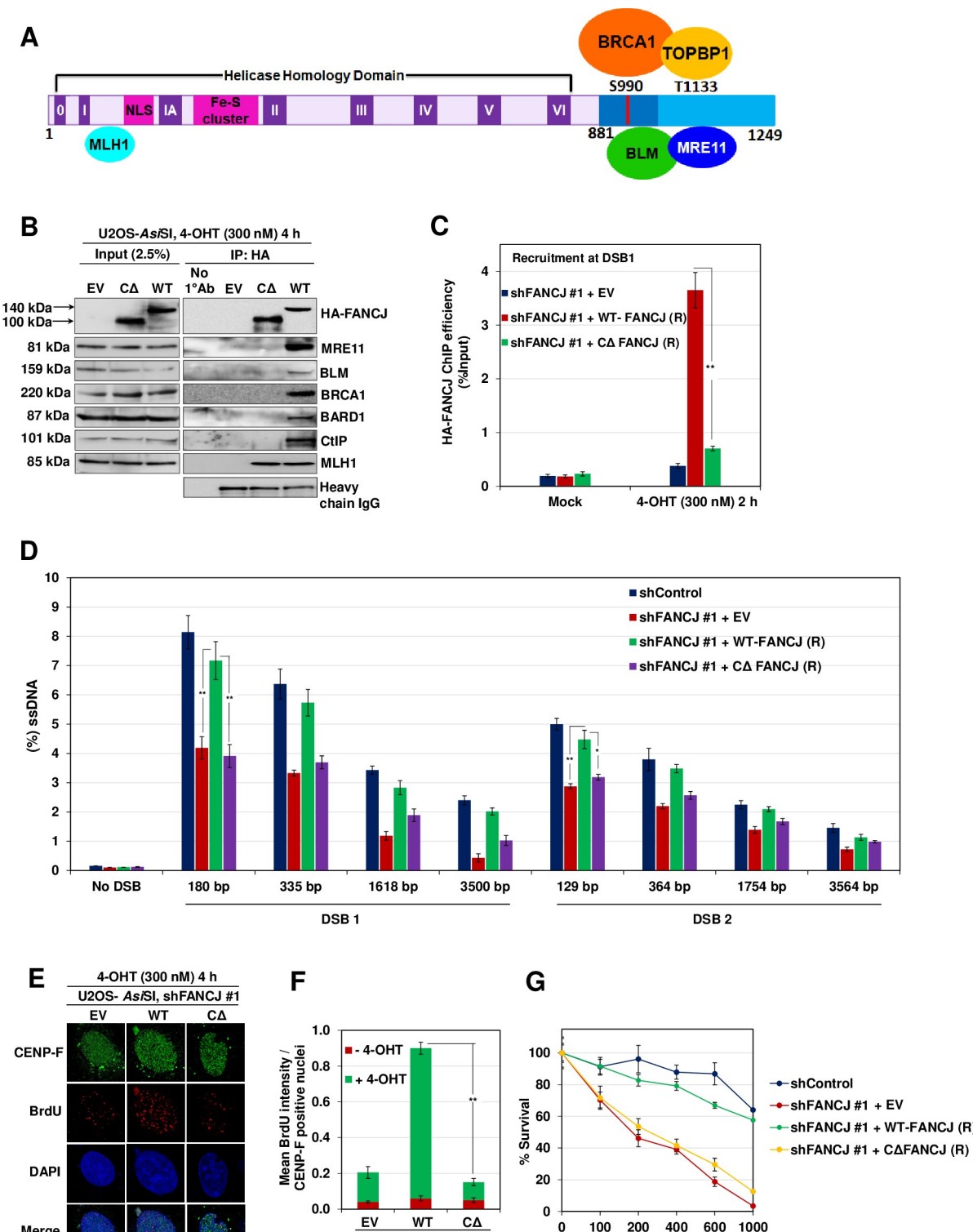

**Fig 3. C-terminus of FANCJ mediates damage induced interaction with CtIP.** (A) Schematic diagram of FANCJ depicting the conserved helicase domains (I–VI), NLS motif, Fe-S cluster, MLH1 binding site in the N-terminus and the C-terminus that is known to bind BRCA1, MRE11, TOPBP1 and BLM. (B) FANCJ depleted ER-*Asi*SI U2OS cells expressing shRNA resistant 1–881 FANCJ-HA-6xHis (CΔ-FANCJ) mutant were treated with 300 nM 4-OHT for 4 h followed by coimmunoprecipitation of the nuclear extracts with antibodies directed against HA-tag. The blot was probed with antibodies against the indicated proteins. (C) FANCJ depleted ER-*Asi*SI U2OS cells expressing shRNA

resistant CΔ-FANCJ mutant were treated with 300 nM 4-OHT for 2 h or mock treated, and ChIP assay was performed using antibody directed against HA-tag. ChIP efficiencies (as percent of input immunoprecipitated) were measured by semiquantitative PCR at 80 bp from *Asi*SI induced DSB1 site. N = 3, with error bars indicating SD and statistical significance was measured by two-tailed *Student*'s t-test of unequal variance. *p < 0.05; **p < 0.01; ***p < 0.001; N.S., non-significant. (D) FANCJ depleted ER-*Asi*SI U2OS cells expressing shRNA resistant CΔ-FANCJ mutant were treated with 300 nM 4-OHT for 4 h or mock treated and measurement of end resection was carried out adjacent to DSB1 and DSB2 sites as previously mentioned. N = 3; with error bars indicating SD and statistical significance was measured by two-tailed *Student*'s t-test of unequal variance. *p < 0.05; **p < 0.01; ***p < 0.001; N.S., non-significant. (E) To measure end resection by BrdU staining, FANCJ depleted ER-*Asi*SI U2OS cells expressing shRNA resistant WT and CΔ-FANCJ-HA-6xHis mutant was pre-labelled with BrdU for 24 h followed by treatment with 300 nM 4-OHT for 4 h or mock treated. Cells were fixed and stained with BrdU antibodies in native conditions to selectively detect ssDNA generated by end resection. CENP-F was used as an S/G2 phase marker to specifically examine DNA end resection during HR. Representative image for BrdU foci is shown. (F) Graph represents mean BrdU intensity from indicated samples in (E). N = 3; error bars indicate standard deviation (SD) and statistical significance was measured by two-tailed *Student*'s t-test of unequal variance. *p < 0.05; **p < 0.01; ***p < 0.001; N.S., non-significant. (G) Survival efficiency of FANCJ depleted ER-*Asi*SI U2OS cells expressing shRNA resistant WT and CΔ-FANCJ mutant in response to *Asi*SI induced breaks.

into the role of FANCJ acetylation in end resection, we generated shRNA resistant FANCJ K1249R acetylation mutant and acetylation mimicking FANCJ K1249Q mutant (S3B Fig). Compared to control cells, FANCJ K1249R was defective in its interaction with CtIP but was competent to bind BRCA1 (Fig 5A) and localize to the sites of DSBs (Fig 5B). However, it was incompetent to support DNA end resection (Fig 5C). Similarly, FANCJ acetylation mutant cells exhibited ~4 fold reduction in BrdU intensity compared to control cells (Fig 5D and 5E). This mutant also showed a reduction in repair efficiency by HR and an increase in NHEJ as well as survival defect (Fig 5F–5H). In contrast, acetylation mimicking FANCJ mutant (K1249Q) was able to interact with CtIP and bind to the DSB sites but showed a moderate defect in DNA end resection (Fig 5A–5C). Consistently, this mutant exhibited significant repair efficiency by HR and cell survival (Fig 5F–5H). These data suggest that the acetylation of FANCJ at K1249 is important for its association with CtIP and for mediating DNA end resection.

## Acetylation of FANCJ is dependent on its phosphorylation at S990

To understand the mechanism underlying FANCJ modification in CtIP interaction and processing of DSBs, we examined whether FANCJ acetylation is dependent on phosphorylation of FANCJ by CDK. We carried out pull-down experiments with cells expressing FANCJ S990A and FANCJ K1249R mutants and analysed FANCJ acetylation in comparison with WT cells. Notably, FANCJ acetylation was abrogated in cells expressing FANCJ S990A mutant (Fig 6A), suggesting that FANCJ phosphorylation is crucial for acetylation of FANCJ, and its interaction with CtIP and ssDNA generation. To investigate whether FANCJ interaction with CtIP is exclusively mediated by acetylation or whether FANCJ phosphorylation is also involved in CtIP interaction, we performed IP experiments with cells expressing FANCJ S990A/K1249Q and FANCJ S990E/K1249R double mutants. Cells expressing FANCJ phosphorylation and acetylation mimicking mutant (S990A/K1249Q) showed interaction with CtIP (Fig 6A). Interestingly, FANCJ phosphomimicking and acetylation deficient (S990E/K1249R) mutant was devoid of its interaction with CtIP (Fig 6A), indicating that FANCJ-CtIP interaction is exclusively dependent on FANCJ acetylation at K1249.

To gain mechanistic insights into CtIP recruitment via FANCJ, we tested the efficiency of CtIP recruitment to the sites of DSBs in cells expressing FANCJ phosphorylation and acetylation mutants and compared it with WT cells. FANCJ S990A mutant which is defective in its interaction with BRCA1 and CtIP showed ~3 fold reduction in CtIP recruitment similar to the FANCJ depleted cells (Fig 6B). In contrast, FANCJ S990E phosphomimetic mutant that is competent in binding to BRCA1 and CtIP exhibited near WT recruitment of CtIP to the DSB1 site. Analysis of FANCJ acetylation mutant (FANCJ K1249R) revealed that this mutant was

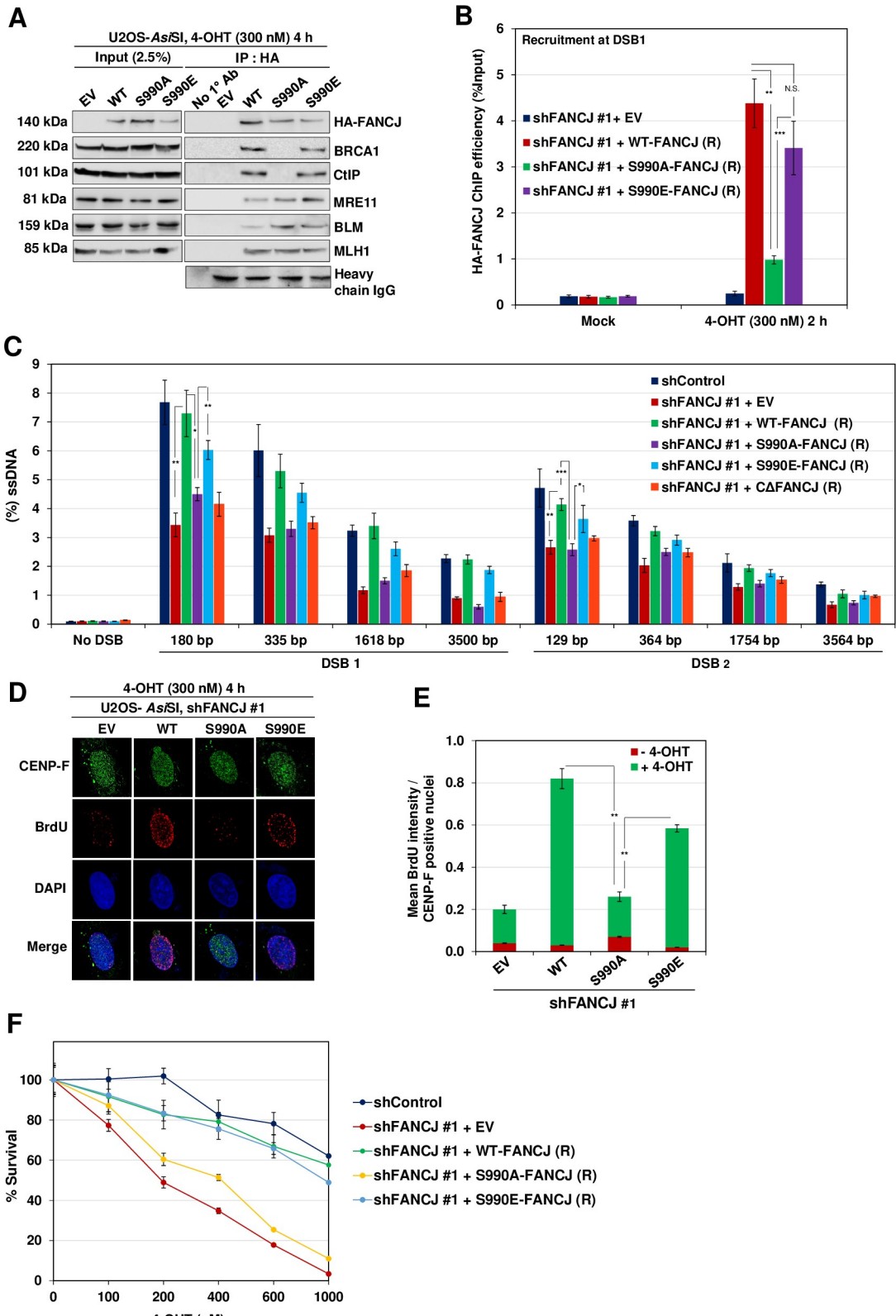

**Fig 4. FANCJ S990 Phosphorylation is essential for CtIP interaction.** (A) FANCJ depleted ER-*Asi*SI U2OS cells expressing shRNA resistant S990A and S990E FANCJ-HA-6x His mutants were treated with 300 nM 4-OHT for 4 h followed by coimmunoprecipitation of the nuclear extracts with antibodies directed against HA-tag. The blot was probed with antibodies

against the indicated proteins. (B) FANCJ depleted ER-*Asi*SI U2OS cells expressing shRNA resistant S990A and S990E FANCJ-HA-6xHis mutants were treated with 300 nM 4-OHT for 2 h or mock treated, and ChIP assay was performed using antibody directed against HA-tag. ChIP efficiencies (as percent of input immunoprecipitated) were measured by semiquantitative PCR at 80 bp from *Asi*SI induced DSB1 site. N = 3, with error bars indicating SD and statistical significance was measured by two-tailed *Student*'s t-test of unequal variance. *p < 0.05; **p < 0.01; ***p < 0.001; N.S., non-significant. (C) FANCJ depleted ER-*Asi*SI U2OS cells expressing shRNA resistant S990A, S990E and CΔ-FANCJ-HA-6xHis mutants were treated with 300 nM 4-OHT for 4 h or mock treated and measurement of end resection was carried out adjacent to DSB1 and DSB2 sites as previously mentioned. N = 3; with error bars indicating SD and statistical significance was measured by two-tailed *Student*'s t-test of unequal variance. *p < 0.05; **p < 0.01; ***p < 0.001; N.S., non-significant. (D) To measure end resection by BrdU staining, FANCJ depleted ER-*Asi*SI U2OS cells expressing shRNA resistant WT, S990A and S990E FANCJ-HA-6xHis mutants were pre-labelled with BrdU for 24 h followed by treatment with 300 nM 4-OHT for 4 h or mock treated. Cells were fixed and stained with BrdU antibodies in native conditions to selectively detect ssDNA generated by end resection. CENP-F was used as an S/G2 phase marker to specifically examine DNA end resection during HR. Representative image for BrdU foci is shown. (E) Graph represents mean BrdU intensity from indicated samples in (D). N = 3; error bars indicate standard deviation (SD) and statistical significance was measured by two-tailed *Student*'s t-test of unequal variance. *p < 0.05; **p < 0.01; ***p < 0.001; N.S., non-significant. (F) Survival efficiency of FANCJ depleted ER-*Asi*SI U2OS cells expressing shRNA resistant WT, S990A and S990E FANCJ mutants in response to *Asi*SI induced breaks.

defective in CtIP loading but acetylation mimicking FANCJ K1249Q mutant was competent to recruit CtIP to a similar extent as that of the WT FANCJ (Fig 6B). These data suggest that both phosphorylation and acetylation of FANCJ plays an important role in CtIP recruitment and end resection.

## BRCA1 is dispensable for interaction of FANCJ with CtIP

A previous study showed that BRCA1 forms a complex with CtIP [40], and FANCJ also interacts with BRCA1 upon phosphorylation at S990 residue [38]. Interestingly, BRCA1 immuno-complex showed interaction with FANCJ and CtIP (Fig 2E). Strikingly, BRCA1-CtIP interaction was not perturbed in FANCJ depleted cells (Fig 7A). However, to investigate whether FANCJ and CtIP interact independently of BRCA1, we carried out IP experiments with CtIP and FANCJ specific antibodies after inducing DSBs by *Asi*SI and analysed the FANCJ-CtIP complexes in control and BRCA1 depleted cells. CtIP immunoprecipitates with FANCJ and this interaction was not perturbed in the BRCA1 depleted cells (Fig 7B). Similarly, compared to control cells, BRCA1 depletion did not affect FANCJ interaction with CtIP (Fig 7B), indicating that FANCJ can efficiently form complex with CtIP in the absence of BRCA1.

CtIP has been shown to bind with BRCT repeats in the C-terminal region of BRCA1 [40]. FANCJ also interacts with BRCA1 BRCT motifs upon phosphorylation at S990 by CDK [38]. Data in Fig 6A clearly shows that CtIP interacts with FANCJ upon acetylation which is, in turn, dependent on phosphorylation of FANCJ at S990 after inducing breaks with *Asi*SI. To investigate whether FANCJ interaction with BRCA1 via S990 phosphorylation of FANCJ is a prerequisite for FANCJ acetylation and CtIP interaction, we carried out our studies with BRCA1 deficient HCC1937 cells. Interestingly, FANCJ was proficient in interacting with CtIP in HCC1937 cells upon induction of DSBs by IR and this association was again dependent on FANCJ S990 phosphorylation and acetylation (Fig 7C). These data suggest that FANCJ interaction with BRCA1 is dispensable for FANCJ acetylation and its association with CtIP.

To investigate whether acetylation dependent interaction of FANCJ with CtIP is DSB specific or FANCJ acetylation can independently mediate CtIP interaction in the absence of DSBs, we carried out IP experiments for FANCJ acetylation mimicking (K1249Q) mutant and studied its interaction with CtIP. Interestingly, this mutant was able to interact with CtIP, suggesting that FANCJ interaction with CtIP is acetylation dependent (Fig 7D). However, FANCJ acetylation was observed upon induction of DSBs but was absent in the undamaged cells (Fig 7D and 7E), implying that DSB induces FANCJ acetylation which is required for CtIP interaction.

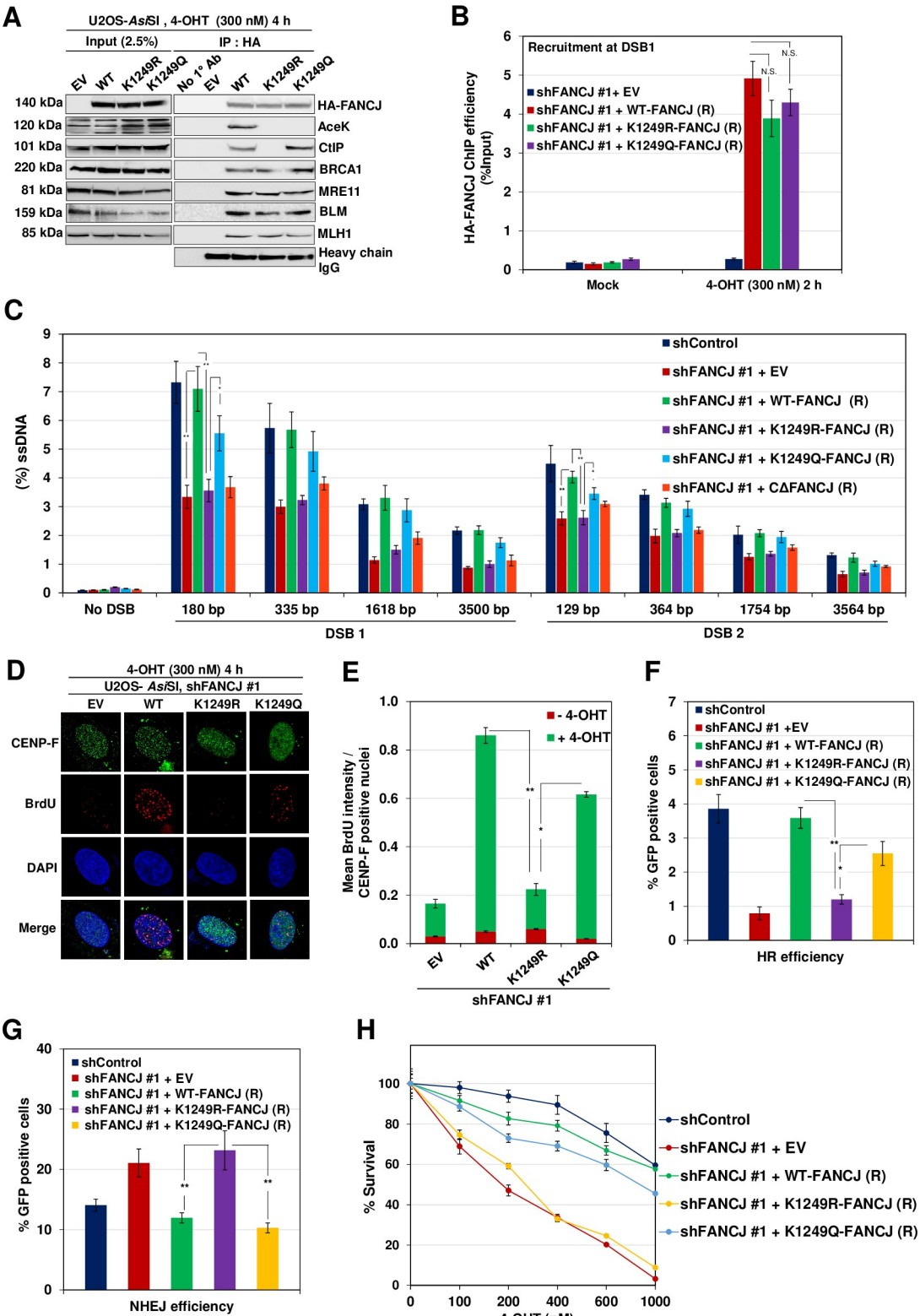

**Fig 5. FANCJ K1249 Acetylation is essential for CtIP interaction.** (A) FANCJ depleted ER-*Asi*SI U2OS cells expressing shRNA resistant K1249R and K1249Q FANCJ-HA-6xHis mutants were treated with 300 nM 4-OHT for 4 h followed by coimmunoprecipitation of the nuclear extracts with antibodies directed against HA-tag. The blot was probed with antibodies against the indicated proteins. (B) FANCJ depleted ER-*Asi*SI U2OS cells expressing shRNA resistant K1249R and K1249Q

FANCJ-HA-6xHis mutants were treated with 300 nM 4-OHT for 2 h or mock treated, and ChIP assay was performed using antibody directed against HA-tag. ChIP efficiencies (as percent of input immunoprecipitated) were measured by semiquantitative PCR at 80 bp from *Asi*SI induced DSB1 site. N = 3, with error bars indicating SD and statistical significance was measured by two-tailed *Student*'s t-test of unequal variance. *p < 0.05; **p < 0.01; ***p < 0.001; N.S., non-significant. (C) FANCJ depleted ER-*Asi*SI U2OS cells expressing shRNA resistant K1249R, K1249Q and CΔ-FANCJ-HA-6xHis mutants were treated with 300 nM 4-OHT for 4 h or mock treated and measurement of end resection was carried out adjacent to DSB1 and DSB2 sites as previously mentioned. N = 3; with error bars indicating SD and statistical significance was measured by two-tailed *Student*'s t-test of unequal variance. *p < 0.05; **p < 0.01; ***p < 0.001; N.S., non-significant. (D) To measure end resection by BrdU staining, FANCJ depleted ER-*Asi*SI U2OS cells expressing shRNA resistant WT, K1249R and K1249Q FANCJ-HA-6xHis mutants were pre-labelled with BrdU for 24 h followed by treatment with 300 nM 4-OHT for 4 h or mock treated. Cells were fixed and stained with BrdU antibodies in native conditions to selectively detect ssDNA generated by end resection. CENP-F was used as an S/G2 phase marker to specifically examine DNA end resection during HR. Representative image for BrdU foci is shown. (E) Graph represents mean BrdU intensity from indicated samples in (D). N = 3; error bars indicate standard deviation (SD) and statistical significance was measured by two-tailed *Student*'s t-test of unequal variance. *p < 0.05; **p < 0.01; ***p < 0.001; N.S., non-significant. (F) I-*Sce*I induced GFP+ frequencies (total GFP; overall HR) in FANCJ depleted U2OS SCR18 cells expressing shRNA resistant WT, K1249R and K1249Q FANCJ-HA-6xHis mutants. *p < 0.05; **p < 0.01; ***p < 0.001; N.S., non-significant. (G) Quantification of NHEJ efficiency in FANCJ depleted U2OS cells expressing shRNA resistant WT, K1249R and K1249Q FANCJ-HA-6xHis mutants transfected with *Hin*dIII linearized NHEJ reporter. N = 3; with error bars indicating SD and statistical significance was measured by two-tailed *Student*'s t-test of unequal variance. *p < 0.05; **p < 0.01; ***p < 0.001; N.S., non-significant. (H) Survival efficiency of FANCJ depleted ER-*Asi*SI U2OS cells expressing shRNA resistant WT, K1249R and K1249Q FANCJ mutants in response to *Asi*SI induced breaks.

## FANCJ promotes end resection in a manner independent of BRCA1-CtIP complex

CtIP has been shown to interact with BRCA1 upon phosphorylation at S327 by CDK2 [41]. To investigate whether FANCJ mediated end resection is independent of the BRCA1-CtIP complex, we generated CtIP S327A phosphodefective mutant and analysed its ability to interact with FANCJ and promote end resection. Consistent with a previous study [41], CtIP S327A mutant was defective with its interaction with BRCA1 (Fig 8A). Interestingly, this mutant was competent to bind with FANCJ and localizing to the sites of DSBs (Fig 8B), and to promote end resection (Fig 8C). However, upon depletion of FANCJ, CtIP phosphomutant was unable to localize to the DSBs and promote end resection (Fig 8B and 8C). These data suggest that FANCJ-CtIP mediated DNA end resection is independent of the BRCA1-CtIP complex.

Although CtIP can independently interact with FANCJ in the absence of BRCA1, FANCJ localization to DSB sites was dependent on BRCA1 (Fig 2A). To test whether BRCA1 and FANCJ participate in the common pathway of DNA end resection, we measured ssDNA generation by depleting BRCA1 and FANCJ individually and compared the DNA end resection upon co-depletion of FANCJ and BRCA1. Depletion of either FANCJ or BRCA1 caused defect in end resection to a similar extent (Fig 8D). However, strikingly, the co-depletion of BRCA1 with FANCJ did not show any further reduction in ssDNA generation (Fig 8D), indicating that FANCJ and BRCA1 participate in the same pathway to mediate DNA end resection.

CtIP T847 is also a phosphorylation target by CDK2 and this phosphorylation is critical for CtIP mediated end resection [42]. To test the role of CtIP T847 phosphorylation in FANCJ mediated end resection, we generated CtIP T847A phosphomutant and investigated its interaction with FANCJ. IP studies revealed that CtIP T847A mutant was able to interact with FANCJ (Fig 8A) and localize to DSBs (Fig 8B) but was defective for promoting end resection (Fig 8C), suggesting that defect associated with CtIP T847A mutant is not due to its inability to interact with FANCJ but likely due to its catalytic inefficiency.

## Helicase activity is important for FANCJ mediated DNA end resection

FANCJ mediated CtIP recruitment and ssDNA generation indicates the structural role of FANCJ in mediating DNA end resection. However, to test whether FANCJ has a catalytic role

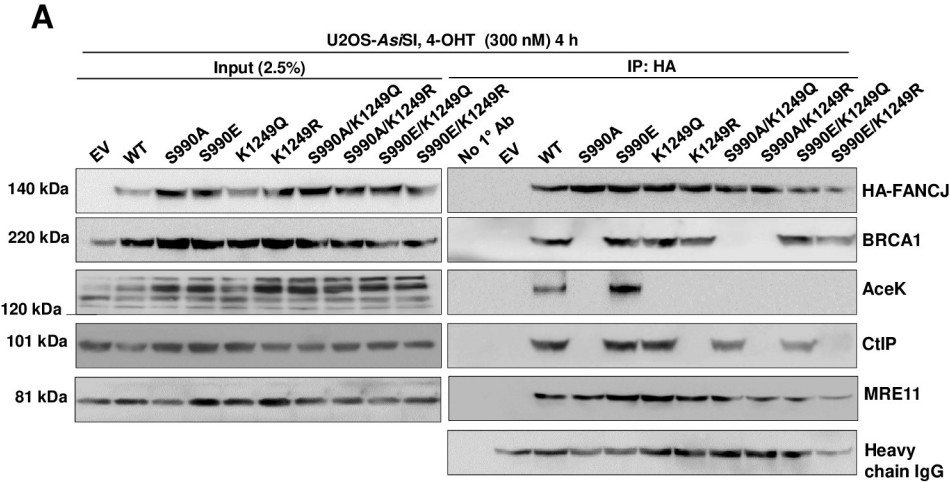

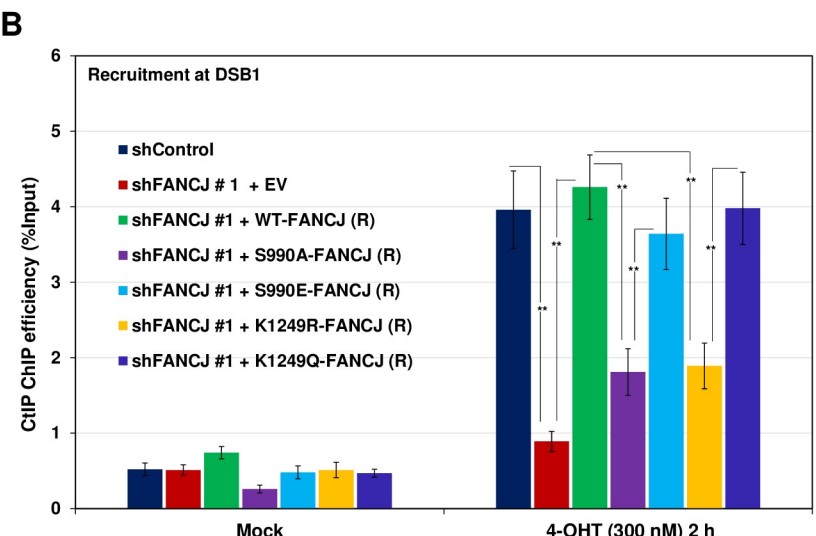

**Fig 6. FANCJ phosphorylation at S990 is a prerequisite for its acetylation at K1249.** (A) Nuclear extracts of FANCJ depleted ER-*Asi*SI U2OS cells expressing FANCJ HA-6xHis single mutants (S990A, S990E, K1249R, K1249Q) and double mutants (S990A/K1249R, S990A/K1249Q, S990E/K1249Q and S990E/K1249R) were co-immunoprecipitated with antibodies against HA-tag after treatment with 300 nM 4-OHT for 4 h. The blot was probed with antibodies against the indicated proteins. (B) FANCJ depleted ER-*Asi*SI U2OS cells expressing shRNA resistant S990A, S990E, K1249R and K1249Q FANCJ-HA-6xHis mutants were treated with 300 nM 4-OHT for 2 h or mock treated, and ChIP assay was performed using antibody directed against CtIP. ChIP efficiencies (as percent of input immunoprecipitated) were measured by semiquantitative PCR at 80 bp from *Asi*SI induced DSB1 site. N = 3, with error bars indicating SD and statistical significance was measured by two-tailed *Student*'s t-test of unequal variance. $^*p < 0.05$; $^{**}p < 0.01$; $^{***}p < 0.001$; N.S., non-significant.

in promoting DNA end resection, we expressed shRNA resistant helicase defective FANCJ K52A and FANCJ K52R mutants and examined CtIP interaction and ssDNA generation (S3C Fig). Consistent with the previous observation, FANCJ K52A and K52R mutants were efficient in their interaction with BRCA1 [19] (Fig 9A). However, interestingly, these mutants were competent in binding to CtIP and localise at the sites of DSBs (Fig 9A and 9B). FANCJ K52 is a Walker A motif lysine residue which is important for ATP binding and hydrolysis [43]. FANCJ K52A is expected to impede both ATP binding and hydrolysis. In contrast, FANCJ K52R is ATP binding competent but is defective for its hydrolysis. An earlier study showed

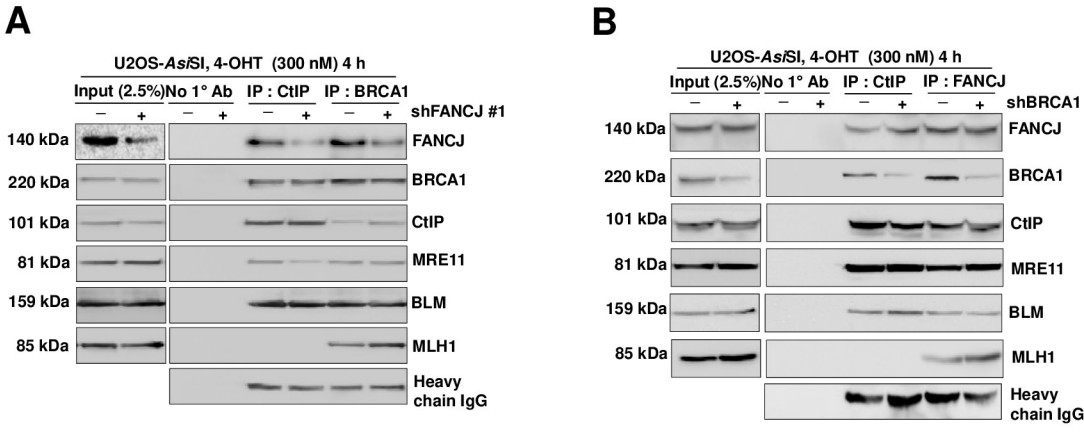

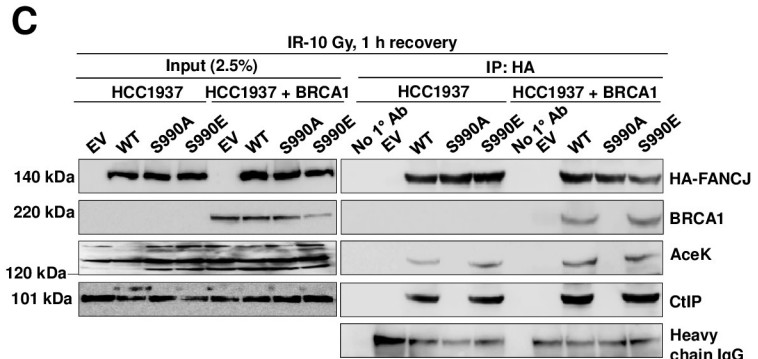

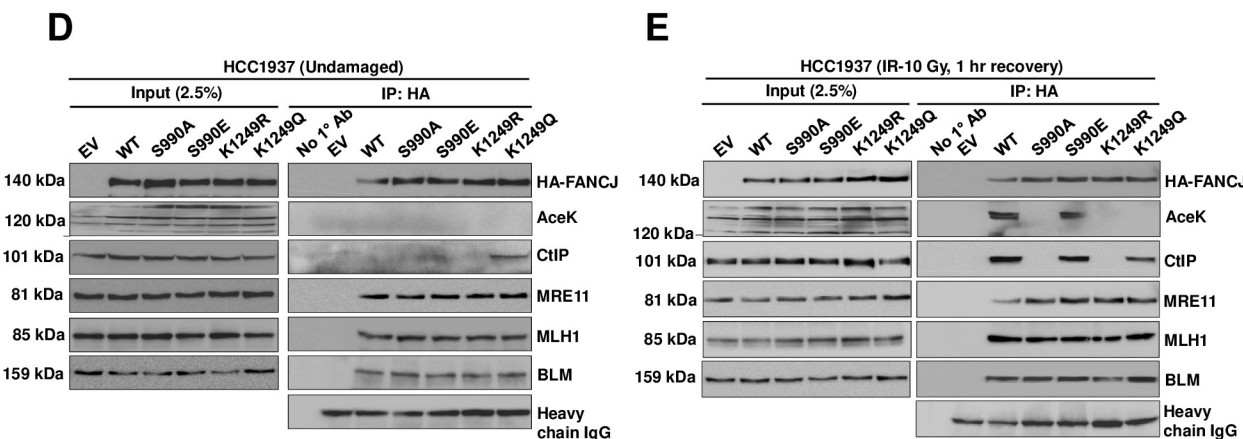

**Fig 7. Interaction of FANCJ with CtIP is independent of BRCA1.** (A) FANCJ depleted ER-*Asi*SI U2OS nuclear extracts were coimmunoprecipitated with antibodies against BRCA1 and CtIP after treatment with 300 nM 4-OHT for 4 h. The blot was probed with antibodies against the indicated proteins. (B) ER-*Asi*SI U2OS cells depleted of BRCA1 were treated with 300 nM 4-OHT for 4 h followed by coimmunoprecipitation of the nuclear extracts with antibodies directed against FANCJ and CtIP. The blot was probed with antibodies against the indicated proteins. (C) FANCJ depleted HCC1937 cells expressing shRNA resistant S990A and S990E FANCJ-HA-6xHis mutants with or without complementation with wt-BRCA1 cDNA were exposed to γ-Irradiation (10 Gy) and recovered for 1 h followed by coimmunoprecipitation of the nuclear extracts with antibodies directed against HA-tag. The blot was probed with antibodies against the indicated proteins. (D) Nuclear extracts of FANCJ depleted HCC1937 cells expressing shRNA resistant S990A, S990E, K1249Q and K1249R FANCJ-HA-6xHis mutants were subjected to co-immunoprecipitation with antibodies against HA-tag. The blot was probed with antibodies against the indicated proteins. (E) FANCJ depleted HCC1937 cells expressing shRNA resistant S990A, S990E, K1249Q and K1249R FANCJ-HA-6xHis mutants were exposed to γ-irradiation (10 Gy) and recovered for 1 h followed by co-immunoprecipitation of the nuclear extracts with antibodies against HA-tag. The blot was probed with antibodies against the indicated proteins.

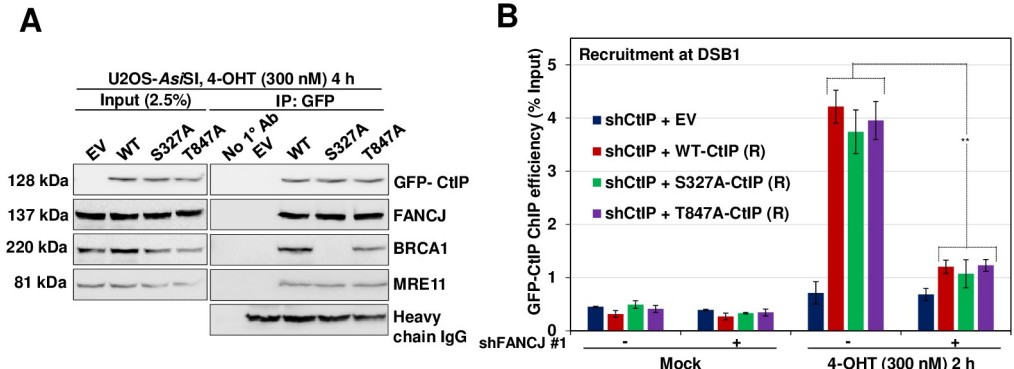

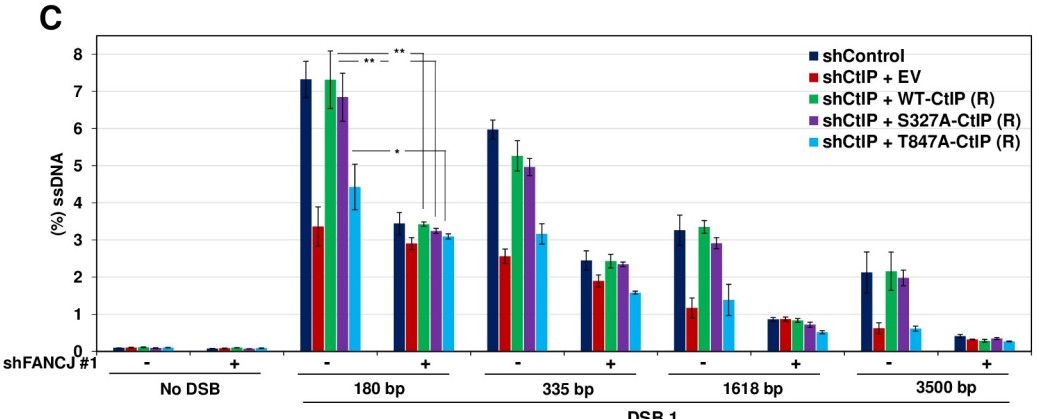

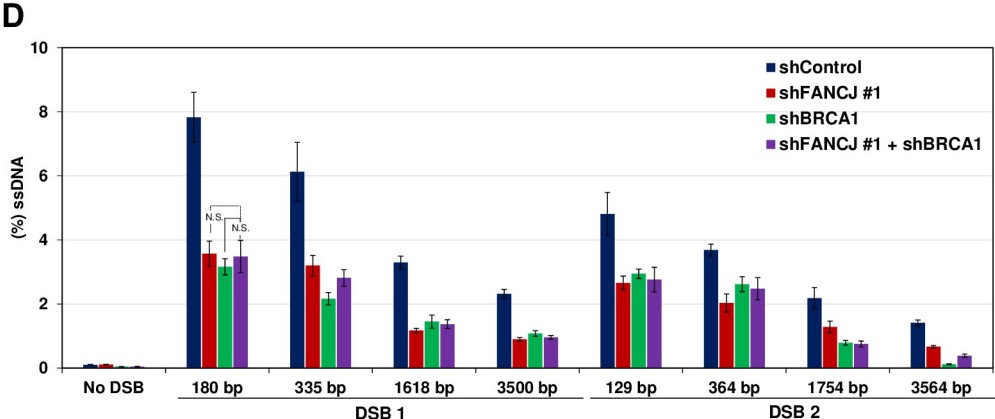

**Fig 8. FANCJ mediated end resection is independent of BRCA1-CtIP complex.** (A) CtIP depleted ER-*Asi*SI U2OS cells expressing shRNA resistant S327A and T847A GFP-CtIP mutants were treated with 300 nM 4-OHT for 4 h followed by coimmunoprecipitation of the nuclear extracts with antibodies directed against GFP-tag. The blot was probed with antibodies against the indicated proteins. (B) CtIP depleted ER-*Asi*SI U2OS cells expressing shRNA resistant S327A and T847A GFP-CtIP mutants were further transfected with shControl or shFANCJ #1 followed by treatment with 300 nM 4-OHT for 2 h or mock treated. ChIP assay was performed using antibody directed against GFP-tag and ChIP efficiencies (as percent of input immunoprecipitated) were measured by semiquantitative PCR at 80 bp from *Asi*SI induced DSB1 site. N = 3, with error bars indicating SD and statistical significance was measured by two-tailed *Student*'s t-test of unequal variance. *p < 0.05; **p < 0.01; ***p < 0.001; N.S., non-significant. (C) CtIP depleted ER-*Asi*SI U2OS cells expressing shRNA resistant S327A and T847A GFP-CtIP mutants were further transfected with shControl or shFANCJ #1 followed by treatment with 300 nM 4-OHT for 4 h or mock treated. Measurement of end resection was carried out adjacent to DSB1 site as previously mentioned. N = 3; with error bars indicating SD and statistical significance was measured by two-tailed *Student*'s t-test of unequal variance. *p < 0.05; **p < 0.01; ***p < 0.001; N.S., non-significant. (D) ER-AsiSI U2OS cells depleted for FANCJ and BRCA1, individually and in combination were treated with 300 nM 4-OHT for 4 h or mock treated and measurement of end resection was carried out adjacent to DSB1 and DSB2 sites as previously mentioned. N = 3; with

error bars indicating SD and statistical significance was measured by two-tailed Student's t-test of unequal variance. $^*$p $<$ 0.05; $^{**}$p $<$ 0.01; $^{***}$p $<$ 0.001; N.S., non-significant.

that FANCJ K52A mutant exhibits more severe defect in HR than ATP binding competent FANCJ K52R mutant [19]. To further understand the ATPase/helicase function of FANCJ in CtIP recruitment and DNA end resection, we analysed the loading of CtIP at the *Asi*SI induced DSB1 site. Strikingly, FANCJ K52A and K52R mutants were proficient in recruiting CtIP to the damaged sites (Fig 9D). Analyses of ssDNA generation at *Asi*SI induced DSB1 and DSB2 sites showed that FANCJ K52A mutant is defective in DNA end resection similar to FANCJ depleted cells (Fig 9E). However, compared to control cells, ATP binding competent FANCJ K52R mutant showed a moderate defect in DNA end resection (Fig 9E). Consistently, FANCJ K52A and K52R mutants showed a severe and moderate defect, respectively in BrdU intensity as well as cell survival in response to *Asi*SI induced breaks (Fig 9F–9H). These data suggest that in addition to having an adaptor role in recruiting CtIP, ATPase/helicase function of FANCJ plays an important role in facilitating DNA end resection.

To gain further insights into whether FANCJ mediated CtIP recruitment alone is sufficient or CtIP recruitment coupled with helicase activity of FANCJ is required for DNA end resection, we analysed CtIP loading at the *Asi*SI induced DSB1 site in cells expressing FANCJ K52A/K1249R double mutant and compared with single mutants. Interestingly, cells expressing FANCJ K52A/K1249R showed a defect in interaction with CtIP and its recruitment similar to FANCJ K1249R single mutant (Fig 9C and 9D). However, consistent with the data presented in Fig 9A, FANCJ K52A mutant was competent to bind and recruit CtIP (Fig 9C and 9D). Analyses of end resection showed that cells expressing FANCJ K52A/K1249R was defective in ssDNA generation similar to the FANCJ K52A mutant (Fig 9E). Together, these data suggest that in addition to FANCJ mediated CtIP recruitment, its helicase activity is critical for promoting DNA end resection.

## Discussion

DNA end resection that occurs in cell cycle specific manner is a prerequisite for the repair of DSBs by HR [44]. Although FANCJ has been shown to play an important role in the repair of DSBs by sister chromatid recombination (SCR) and in the suppression of SCR associated gene amplification [19], the precise mechanism by which FANCJ regulates HR is elusive. Here, we find that FANCJ participates in DNA end resection by recruiting CtIP to the sites of DSBs. The association of CtIP with FANCJ is dependent on FANCJ K1249 acetylation which is mediated by CDK dependent phosphorylation of FANCJ. However, FANCJ interaction with BRCA1 is dispensable for FANCJ acetylation and its interaction with CtIP. Notably, FANCJ promotes end resection in a manner independent of the BRCA1-CtIP complex. In addition to the scaffolding role, FANCJ helicase activity is also important for DNA end resection. Together, our work identifies a novel function of FANCJ helicase in DNA end resection and provides mechanistic basis of FANCJ as an important regulator of DSB processing during HR repair.

Similar to canonical DNA end resection factors such as MRE11, CtIP, and BLM [11, 12], FANCJ depleted cells showed dysregulated ssDNA formation after *Asi*SI induced breaks. This observation was further corroborated by quantitative measurement of DNA end resection at two DSB loci in chromosome 1. FANCJ deficient cells exhibited a 2–3 fold reduction in DNA end resection that was measured in the range of 130bp to 3.5 kb, clearly indicating the role of FANCJ in DSB processing. Initiation of end resection occurs by the assembly of multiple factors such as MRE11, CtIP, EXO1, DNA2 and BLM to the sites of DSBs [12]. Chromatin IP

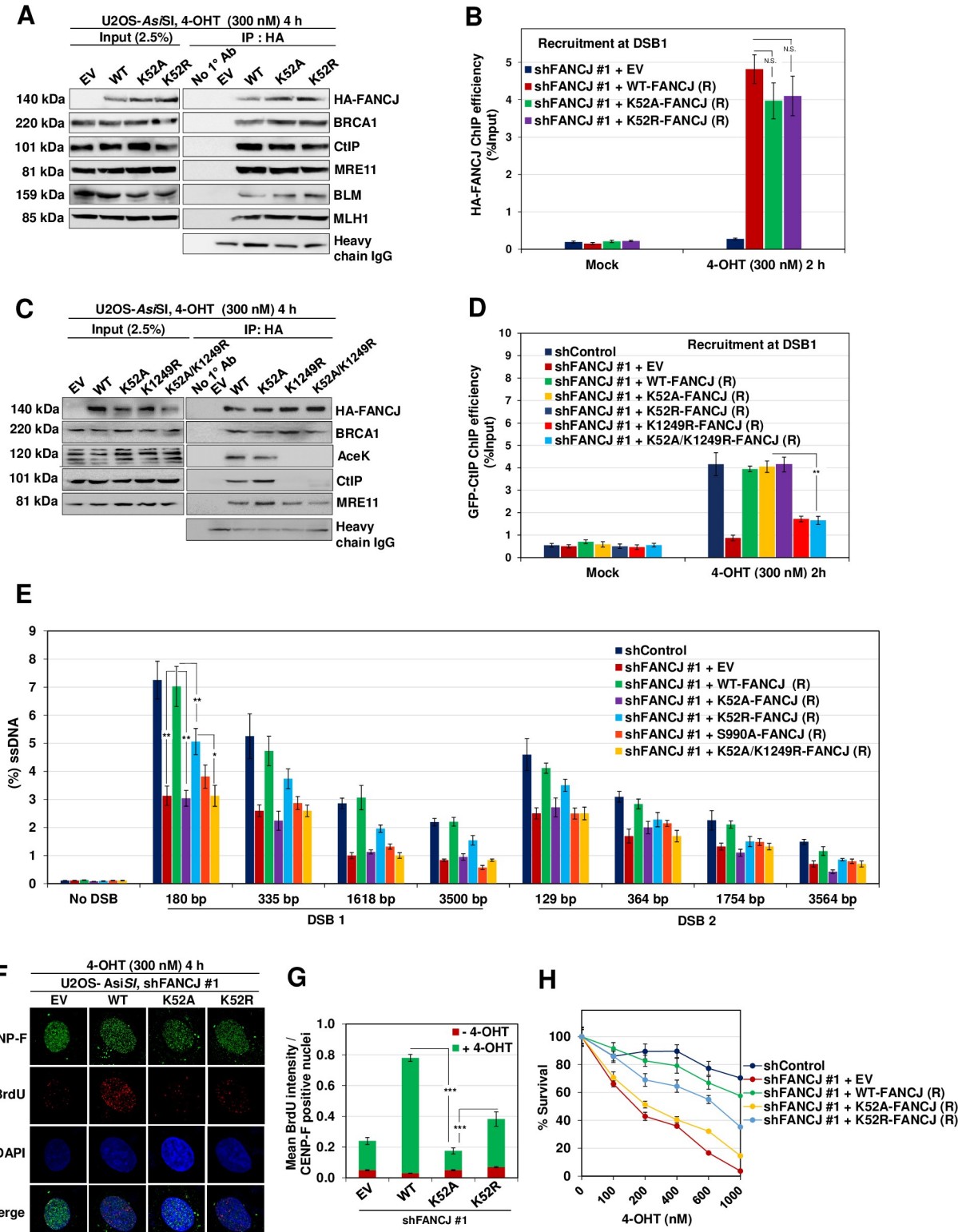

**Fig 9. FANCJ ATPase/Helicase activity is essential for DNA end resection.** (A) FANCJ depleted ER-*Asi*SI U2OS cells expressing shRNA resistant K52A and K52R FANCJ-HA-6xHis mutants were treated with 300 nM 4-OHT for 4 h followed by coimmunoprecipitation of the nuclear extracts with antibodies directed against HA-tag. The blot was probed with antibodies against the indicated proteins. (B) FANCJ depleted ER-*Asi*SI U2OS cells expressing shRNA resistant K52A and K52R FANCJ-HA-6xHis mutants were treated with 300 nM 4-OHT for 2 h or mock treated, and ChIP assay was performed using antibody directed against HA-tag. ChIP efficiencies (as percent of input immunoprecipitated) were

measured by semiquantitative PCR at 80 bp from *Asi*SI induced DSB1 site. N = 3, with error bars indicating SD and statistical significance was measured by two-tailed *Student*'s t-test of unequal variance. *p < 0.05; **p < 0.01; ***p < 0.001; N.S., non-significant. (C) Nuclear extracts of FANCJ depleted ER-*Asi*SI U2OS cells expressing shRNA resistant K52A, K1249R and K52A/K1249R FANCJ-HA-6xHis mutants were subjected to co-immunoprecipitation with antibodies against HA-tag. The blot was probed with antibodies against the indicated proteins. (D) FANCJ depleted ER-*Asi*SI U2OS cells expressing shRNA resistant K52A, K52R, K1249R and K52A/K1249R FANCJ-HA-6xHis mutants were treated with 300 nM 4-OHT for 2 h or mock treated, and ChIP assay was performed using antibody against CtIP. ChIP efficiencies (as percent of input immunoprecipitated) were measured by semiquantitative PCR at 80 bp from *Asi*SI induced DSB1 site. N = 3, with error bars indicating SD and statistical significance was measured by two-tailed *Student*'s t-test of unequal variance. *p < 0.05; **p < 0.01; ***p < 0.001; N.S., non-significant. (E) FANCJ depleted ER-*Asi*SI U2OS cells expressing shRNA resistant K52A, K52R, S990A and K52A/K1249R FANCJ-HA-6xHis mutants were treated with 300 nM 4-OHT for 4 h or mock treated and measurement of end resection was carried out adjacent to DSB1 and DSB2 sites as previously mentioned. N = 3; with error bars indicating SD and statistical significance was measured by two-tailed *Student*'s t-test of unequal variance. *p < 0.05; **p < 0.01; ***p < 0.001; N.S., non-significant. (F) To measure end resection by BrdU staining, FANCJ depleted ER-*Asi*SI U2OS cells expressing shRNA resistant WT, K52A and K52R FANCJ-HA-6xHis mutants were pre-labelled with BrdU for 24 h followed by treatment with 300 nM 4-OHT for 4 h or mock treated. Cells were fixed and stained with BrdU antibodies in native conditions to selectively detect ssDNA generated by end resection. CENP-F was used as an S/G2 phase marker to specifically examine DNA end resection during HR. Representative image for BrdU foci is shown. (G) Graph represents mean BrdU intensity from indicated samples in (F). N = 3; error bars indicate standard deviation (SD) and statistical significance was measured by two-tailed *Student*'s t-test of unequal variance. *p < 0.05; **p < 0.01; ***p < 0.001; N.S., non-significant. (H) Survival efficiency of FANCJ depleted ER-*Asi*SI U2OS cells expressing shRNA resistant WT, K52A and K52R FANCJ mutants in response to *Asi*SI induced breaks.

studies reveal damage specific enrichment of FANCJ at DSB sites similar to MRE11 and CtIP, and this recruitment was affected by BRCA1 depletion and to a lesser extent by MRE11 depletion but not by CtIP or BLM. Indeed, FANCJ localization to the DSB sites but not to ICL lesions has been shown to be dependent on MRE11 [33]. In agreement with the previous study [33], MRE11 binds to the C-terminal region of FANCJ. It is likely that MRE11 localization to the DSB sites recruits FANCJ via its interaction at the C-terminus of FANCJ.

CtIP and its functional orthologs in various organisms play a key role in DNA end resection by binding to DNA ends and serving as a cofactor in MRE11 mediated end resection [45, 46]. MRE11, BRCA1 tumor suppressor, and BLM helicase interact with the C-terminus of FANCJ [16]. Interestingly, we find that CtIP also binds to the C-terminal region of FANCJ upon DSB induction, and deletion of 368 residues from the C-terminal end of FANCJ affected its physical association with CtIP, recruitment to DSB sites and end resection ability to a similar extent as that of FANCJ deficient cells. FANCJ S990 is a phosphorylation target by CDK and this phosphorylation is crucial for its interaction with BRCA1 at damaged sites [38]. Expression of FANCJ S990A mutant abrogated CtIP interaction but not MRE11 and BLM. Moreover, this mutant failed to assemble at damaged sites and promote end resection. However, this deficiency was rescued by the expression of FANCJ S990E phosphomimetic mutant, indicating that FANCJ S990 phosphorylation is critical for FANCJ recruitment to DSB sites and interaction with CtIP to facilitate DNA end resection (Fig 10).

FANCJ K1249 has been shown to be an acetylation target by CBP and this modification is required for FANCJ mediated DNA damage responses [39]. Our data shows that FANCJ acetylation is critical for CtIP interaction but not for its association with MRE11, BRCA1, and BLM. Interestingly, FANCJ acetylation defective mutant was competent to bind to damaged sites but was defective with end resection, implying that this mutant was competent for phosphorylation by CDK and its interaction with BRCA1. Notably, FANCJ S990A mutant was defective for acetylation and thereby affected FANCJ interaction with CtIP and FANCJ mediated end resection (Fig 10). Our analysis with BRCA1 deficient HCC1937 cells revealed that FANCJ interaction with BRCA1 is not critical for K1249 acetylation and its association with CtIP. These data clearly suggest that FANCJ S990 phosphorylation by CDK mediates acetylation at K1249 in response to DSBs (Fig 10). However, IP studies with FANCJ double mutants indicate that the physical interaction of CtIP with FANCJ is exclusively dependent on its acetylation but not by phosphorylation. However, further studies are required to understand the

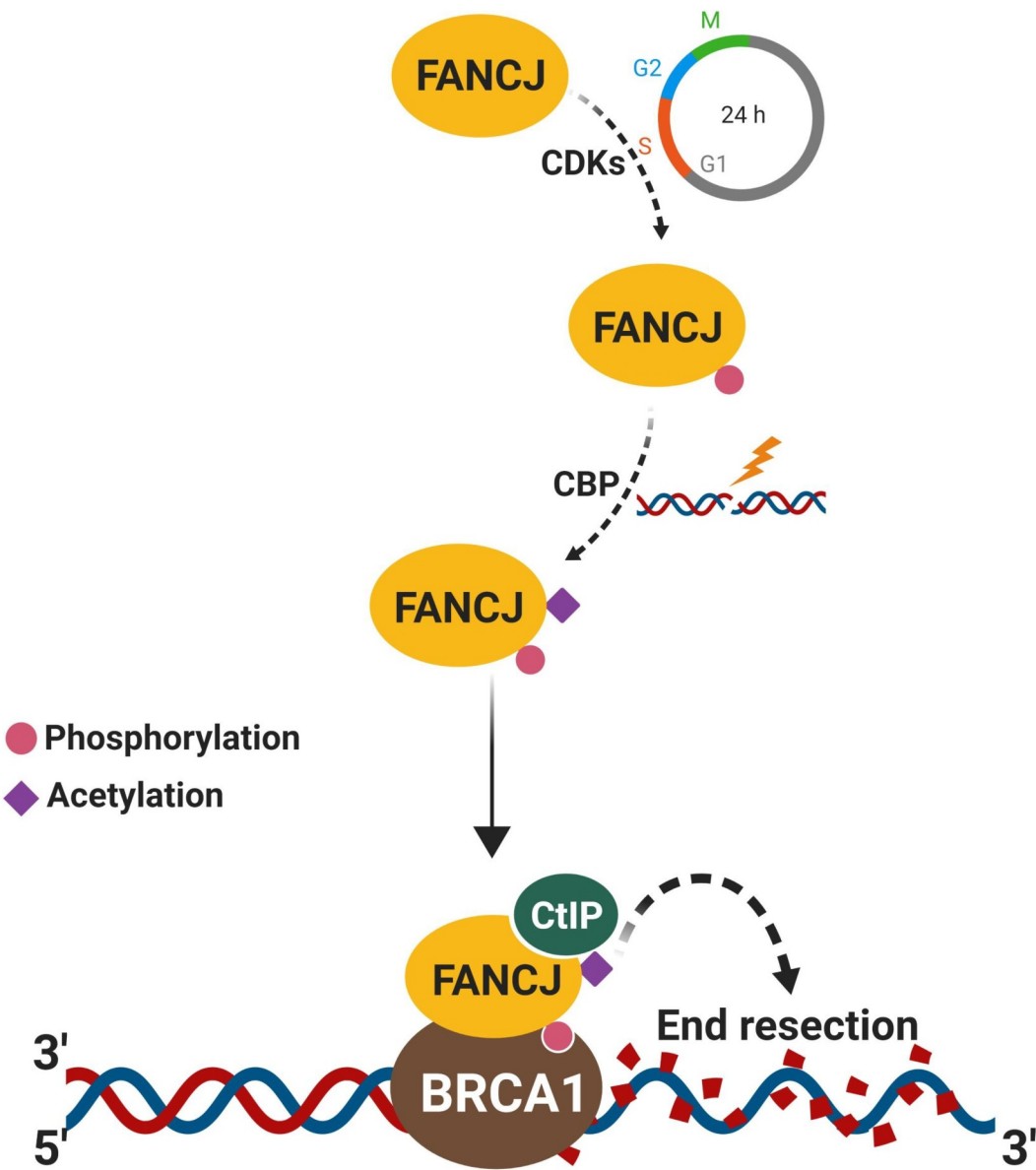

**Fig 10. A model to depict FANCJ mediated DNA end resection.** CDK catalyzes FANCJ S990 phosphorylation during S-phase which facilitates FANCJ acetylation and CtIP interaction in a DSB specific manner. FANCJ-CtIP complexes are recruited to DSBs via BRCA1which is mediated by CDK dependent phosphorylation of FANCJ at S990. Loading of FANCJ-CtIP complexes to DSBs drives DNA end resection and promotes HR.

molecular mechanism underlying phosphorylation-dependent acetylation of FANCJ and its interaction with CtIP to promote DNA end resection.

In addition to canonical players, recent studies show that RECQL4 [47], SAMHD1 [48], AND-1 [49], CTCF [50] and AUNIP [51] participates in DNA end resection by facilitating CtIP recruitment to the DSB sites. Our data demonstrating FANCJ dependent CtIP recruitment mediated by phosphorylation and acetylation identifies a novel mechanism by which DSB ends are processed in mammalian cells for repair by HR (Fig 10). Protein interactions that occur through acetylation modification require bromodomain which is absent in CtIP. The underlying mechanism by which FANCJ acetylation mediates CtIP interaction needs

further investigation. Moreover, further studies are required to understand whether an interplay exists between FANCJ and other factors in recruiting CtIP to the sites of DSBs to promote DNA end resection.

CDK plays an important role in determining the pathway choice of DSB repair by HR and also regulating the end resection machinery to effectively generate ssDNA for initiating HR [44]. This is achieved by CDK mediated phosphorylation of CtIP at S327 which is required for 53BP1 displacement from sites of DSB ends and channelizing DSBs to HR mediated repair [27]. CDK also facilitates CtIP phosphorylation at T847 during S/G2 phase to promote DNA end resection [27, 52]. EXO1 mediated long-range resection of DNA ends has been shown to be regulated by CDK dependent phosphorylation of EXO1 at multiple sites [53]. In addition, CDK also phosphorylates RECQL4 to stimulate its helicase activity and promote DNA end resection [54]. Our study extends this phenomenon of CDK dependent regulation of DNA end resection by phosphorylating FANCJ at S990 which is essential for FANCJ acetylation, which in turn is required for CtIP recruitment (Fig 10).

BRCA1 has been shown to interact with CtIP upon phosphorylation by CDK at S327 and this interaction was believed to play a role in DNA end resection [55, 56]. We find that CtIP S327A mutant devoid of BRCA1 binding was competent to bind FANCJ and promote end resection, suggesting that FANCJ mediated end resection is independent of BRCA1-CtIP complex. Indeed, recent studies clearly demonstrate that BRCA1-CtIP interaction mediated by phosphorylation of CtIP S327 is not essential for DNA end resection [57–59]. The tumor suppressor BRCA1 plays an important role in determining the pathway choice of repair by participating in end resection and promoting HR mediated DSB repair [27, 60, 61]. BRCA1 exists in multiple complexes: BRCA1-A with abraxas; BRCA1-B with FANCJ and BRCA1-C with CtIP [60, 62, 63]. The fact that FANCJ recruitment to damaged sites is dependent on BRCA1 via FANCJ S990 phosphorylation, clearly indicates that FANCJ-CtIP mediated end resection is dependent on BRCA1 but independent of BRCA1-CtIP complex.

MRN complex in addition to having catalytic roles in DNA end resection also serves as a structural component in recruiting DNA end resection factors and stimulating their activity [11]. Our analysis with FANCJ Walker A motif lysine mutants revealed interesting findings. Although the FANCJ K52A ATP binding deficient mutant was competent to bind and recruit CtIP to DSB sites, it was devoid of end resection activity. In contrast, ATP binding competent but hydrolysis deficient FANCJ K52R mutant was partially functional in promoting DNA end resection, implying that in addition to its scaffolding role in CtIP recruitment, FANCJ helicase activity is required for resection of DNA ends. Indeed, studies show that FANCJ K52R mutant is defective for ATP hydrolysis, DNA unwinding [64, 65] and HR [19]. Moreover, it has been shown that ATP binding induced conformation change coupled with ATP hydrolysis is essential for the helicase activity of RECQ1[66] and YxiN [67]. The defective end resection with FANCJ K52R mutant implies that ATP binding induced conformational changes coupled with the motor activity of FANCJ is critical for efficient end resection. FANCJ is known to unwind a variety of DNA substrates including forked duplex, flap structures, D-loop and G4 DNA structures [16]. Conceivably, these activities may facilitate resolving secondary structures near the DSB ends or while processing the DNA ends. However, further studies are required to understand how FANCJ helicase activity regulates DNA end resection.

Our previous study showed that FANCJ depleted cells exhibit a reduction in overall HR and a bias towards long-tract gene conversions (LTGC) [19]. HR requires resection of DNA ends for the proper loading and assembly of RAD51 nucleofilaments [13]. The current study provides evidence for the role of FANCJ in DNA end resection, and defect in end resection could account for the impaired HR in FANCJ deficient cells. Defect in the end resection might also influence the outcome of HR. SDSA mechanism of DSB repair involves displacement of

the nascent strand from the D-loop and annealing to the second end of broken chromosome [13, 68]. Precise annealing of nascent DNA to the second end of the broken chromosome requires sufficiently resected DNA ends. Defect in the end resection could lead to reinvasion of the nascent strand to the sister chromatid and could account for the LTGC associated gene amplifications/duplications [19]. Interestingly, a chromatin remodelling factor ATRX has been shown to be required for extended DNA repair synthesis during HR [69]. It is unclear whether ATRX plays a role in DNA end resection which might influence HR and its outcome.

## Materials and methods

### Cell lines

Human cell lines U2OS and U2OS-SCR18, ER-*Asi*SI U2OS and HCC1937 were kind gifts from Ralph Scully (Harvard Medical School, Boston, USA), Gaelle Legube (CNRS Toulouse, France) and Priya Srinivas (RGCB, Kerala, India), respectively. These cells were grown in Dulbecco's modified Eagle's medium (DMEM) supplemented with 10% FBS and penicillin/streptomycin (Sigma-Aldrich) at 37˚C in humidified air containing 5% CO$_2$. U2OS-SCR18 and ER-*Asi*SI U2OS cells were cultured under Puromycin selection (2 mg/ml; Sigma-Aldrich).

### DNA constructs and transfections

The pcDNA3-myc-His FANCJ cDNA was a kind gift from Sharon Cantor (UMass Medical School). As described previously [19], hemagglutinin (HA)-6xHis-tagged human wild-type (WT) FANCJ and its mutant constructs were generated by PCR-based mutagenesis using primer sequences indicated in S3 Table and cloned into the modified pcDNA3β vector using *Eco*RV and *Xho*I restriction sites. The FANCJ shRNA resistant WT and mutant constructs were designed by introducing silence mutations in the FANCJ cDNA sequence corresponding to the FANCJ shRNA#1 sequence (S4 Table). These primers were used for restriction-free cloning with PfuTurbo polymerase. The design and construction of the SCR reporter and I-*Sce*I expression vector have been described previously [70]. All short hairpin RNA (shRNA) constructs were generated by using previously reported small interfering RNA (siRNA) sequences (S4 Table) and cloned into the pRS shRNA vector. The pCW-GFP-CtIP was purchased from Addgene (Plasmid # 71109). The CtIP shRNA resistant mutants S327A and T847A were generated by PCR-based mutagenesis. The complementation of HCC1937 cells was carried out with pDEST-mCherry-LacR-BRCA1 purchased from Addgene (Plasmid # 71115). All plasmid transfections were carried out by electroporation using a Bio-Rad gene pulsar X cell (250 V and 950 μF). After 24 h of transfections with shRNA constructs, cells were transfected with shRNA resistant plasmids. Cells were incubated with 4-OHT 16 h post transfections with shRNA resistant constructs. After 4 h of incubation with 4-OHT, cells were processed for end resection assay. For ChIP analysis, cells were processed 2 h post incubation with 4-OHT.

### Immunostaining

Exponentially growing U2OS cells were seeded onto sterile coverslips. To detect ssDNA, cells were pre-labelled with 10 μM BrdU for 24 h before the transfection with shRNA plasmids. Where appropriate, cells were treated with 300 nM 4-Hydroxytamoxifen;4-OHT (Sigma) for 4 h. After treatment, the cells were washed with PBS. Pre-extraction was performed using 0.2% Triton X-100 in PBS on ice for 1 min for BrdU staining. Cells were fixed in 3.7% formaldehyde for 10 min at room temperature followed by permeabilization with 0.2% Triton X-100 in PBS for pRPA2(S4/S8) and γ-H2AX staining. Later, cells were blocked in 0.5% BSA/0.2% TritonX-

100 for 30 min. The cells were then incubated with the indicated primary antibodies (S5 Table) and FITC/TRITC-conjugated secondary antibodies (1:100) (Sigma) for 1 h each at room temperature, and then stained with DAPI before mounting onto slides. Images were acquired using Olympus confocal microscope FV3000 and processed using Olympus fluoview image browser software.

### Recombination assays

HR assays were performed as described previously [71]. In brief, U2OS SCR18 cells were transfected with appropriate shRNA constructs. After 24 h, $2 \times 10^6$ cells were transfected with 24 μg of I-*Sce*I expression plasmid. 48 h later, GFP+ cells were scored by FACS analysis using BD biosciences Verse flow cytometer. In each experiment, the percentage of GFP positive cells was measured in triplicate samples, and I-*Sce*I-transfected values were corrected for transfection efficiency (~60–70%). The spontaneous GFP+ frequency (<0.01%) was subtracted from this value to obtain the I-*Sce*I-induced GFP+ frequency. Data represent the mean of at least three independent experiments with SD values indicated by error bars.

### NHEJ assay

NHEJ reporter assay was performed as described previously [72, 73]. In brief, U2OS cells were transfected with appropriate shRNA constructs. Pem1-Ad2-EGFP reporter was linearized using either *Hin*dIII or I-*Sce*I. After 24 h of transfection with respective shRNA constructs, 7 μg of the linearized reporter was co-transfected with 7 μg of mCherry plasmid by electroporation. Cells were harvested for analysis by flow cytometry after 48h of incubation using flow cytometer verse (BD Biosciences). Results are represented as a ratio of double-positive cells (EGFP+/Cherry+) to the total number of Cherry-positive cells to demonstrate NHEJ efficiency.

### Cell synchronization and cell cycle analysis

Cell synchronization and cell cycle analysis were performed as described previously [74]. ssDNA generation was measured by quantitative PCR, and BrdU foci formation in cells synchronized in S/G2 phase. Cells were arrested at the G2/M phase by the addition of RO-3306 CDK1 inhibitor (10 μM, 16 h). The floating mitotic cells were then collected by shaking-off, washed with fresh media and then re-plated. The cells collected after 12 h were predominantly in S/G2 phase as analysed by flow cytometry (S1B Fig). Briefly, collected single-cell suspensions were fixed overnight with 70% ethanol in PBS at -20˚C. After centrifugation, the cells were incubated with RNaseA (0.1 mg/ml) in PBS at 42˚C for 4 h and then incubated for 10 min with 50 μg/ml propidium iodide (PI) in dark. A total of $1 \times 10^4$ cells were analysed by Verse flow cytometer (BD Biosciences). Aggregates were gated out and the percentage of cells with 2N and 4N DNA content were calculated using FACSDiva Version 6.1.1 software (BD Biosciences).

### Genomic DNA extraction

ER-*Asi*SI U2OS cells synchronised in S/G2 phase were treated with 4-Hydroxytamoxifen; 4OHT (Sigma) for a maximum period of 4 h. After harvesting, the cell suspension was centrifuged and resuspended with 37˚C 0.6% low-gelling point agarose in PBS at a density of $6 \times 10^6$ cells/ml. A 50 μl cell suspension was dropped onto a piece of Parafilm placed on ice to generate a solidified agar ball, which was then transferred to a 1.5 ml eppendorf tube. The agar ball was incubated with 1 ml of ESP buffer (0.5M EDTA, 2% N-lauroylsarcosine, 1 mg/ml proteinase-

K, 1mM $CaCl_2$, pH 8.0) for 20 h at 16˚C with slow rotation, followed by treatment with 1 ml of HS buffer (1.85 M NaCl, 0.15 M KCl, 5 mM $MgCl_2$, 2 mM EDTA, 4 mM Tris, 0.5% Triton X-100, pH 7.5) for 20 h at 16˚C with slow rotation. This was followed by repeated washes with 1 ml of phosphate buffer (8 mM $Na_2HPO_4$, 1.5 mM $KH_2PO_4$, 133 mM KCl, 0.8 mM $MgCl_2$, pH 7.4) for 6x1 h at 4˚C with slow rotation. The agar ball was melted by placing the tube in a 70˚C heat block for 10 min. The melted sample was diluted 15-fold with 70˚C $ddH_2O$, mixed with an equal volume of appropriate 2x NEB restriction enzyme buffer and stored at 4˚C for further use.

## Quantitative PCR for end resection analysis

End resection was assayed in ER-*Asi*SI cells that were predominantly in S/G2 phase by cell synchronization procedure mentioned earlier in the Methods section. The extent of resection adjacent to specific DSBs was measured by quantitative polymerase chain reaction (qPCR) as described previously [24]. The sequences of qPCR primers are shown in S1 Table. 20μL of genomic DNA sample (~200 ng in 1x CutSmart NEB restriction enzyme buffer) was digested or mock digested with 20 units of restriction enzymes (*NmeA*III, *Ava*I, *BsrG*I, *Bam*HI-HF or *Hin*dIII-HF; New England Biolabs) at 37˚C overnight. 5 μl of digested or mock digested samples (~20 ng) were used as templates in 20 μl of qPCR reaction containing 10 μl of 2x iTaqUniversal SYBR Green Supermix, 500 nM of each primer using iCycler iQReal-Time PCR (Bio-Rad). The % ssDNA generated by resection at selected sites was determined as previously described [24]. Briefly, for each sample, a ΔCt was calculated by subtracting the Ct value of the mock-digested sample from the Ct value of the digested sample. The % ssDNA was calculated using algorithm: ssDNA% = $1/(2^{\wedge}(\Delta Ct-1) + 0.5)*100$ [24]. Data represent the mean of at least three independent experiments with SD values indicated by error bars.

## Immunoprecipitation

Immunoprecipitations were carried out as described previously [75]. After harvesting the cells, the cytosolic protein fraction was removed by incubation in hypotonic buffer (10 mM HEPES, pH 7, 50 mM NaCl, 0.3M sucrose, 0.5% Triton X-100, supplemented with protease inhibitor (Roche), 20 mM NaF, 10 mM Trichostatin (TSA) and 5 mM nicotinamide (NAM) for 15 min on ice and centrifuged at 1500xg for 5 min. The remaining pellet was resuspended in lysis buffer (20 mM Tris-HCl (pH 7.5), 100 mM NaCl, 1% Nonidet P-40 (NP-40), 5% glycerol, 1 mM EDTA, 1 mM $MgCl_2$, 1 mM ATP, 1 mM dithiothreitol (DTT), 20 mM NaF, protease inhibitor (Roche), 10 mM TSA, 5 mM NAM and Benzonase (10 U/μl). After sonication at low amplitude, lysates were cleared by centrifugation for 10 min. Where appropriate, 1 mg of lysates was incubated with 6 μg of the indicated antibodies for 12–16 h at 4˚C. Lysates were then incubated with 30 μl of Protein A/G beads (GE Healthcare) for 6 h at 4˚C. Ig–antigen complexes were washed extensively and eluted in 2x Laemmli sample buffer at 90˚C with shaking for 45 min before SDS-PAGE.

## Chromatin Immunoprecipitation

Cells were cultured overnight at a density of $1\times10^7$ per 150 mm petri dish and subjected to treatment with either solvent or 300 nM 4-OHT for 2 h, followed by incubating cells in 1% formaldehyde for 15 min at room temperature, and the reaction was stopped by 10 min incubation with 125 mM glycine. Cells were collected and washed three times by PBS, and soluble proteins were removed by incubation of cells with 0.5% Triton X-100 in PBS supplemented with 10 mM NaF, 10 mM TSA and 5 mM NAM for 5 min on ice. The remaining pellet was lysed by lysis buffer (50 mM Tris-HCl (pH 8.0), 10 mM EDTA (pH 8.0), 0.1% SDS and

protease inhibitor cocktail (Roche), 10 mM NaF, 10 mM TSA and 5 mM NAM) on ice for 10 min followed by sonication (30 s-on/30 s-off, eight times) at 20% amplitude using Q-sonica model Q-500. Lysates were diluted using dilution buffer (50 mM Tris-HCl (pH 8.0), 150 mM NaCl, 1% Triton X-100, 0.1% SDS, protease inhibitor cocktail, 10 mM NaF, 10 mM TSA and 5 mM NAM). Where appropriate, for each sample, 1 mg of lysates was incubated with 6 μg of the indicated antibodies for 12–16 h at 4˚C. Antibody-chromatin complexes were pulled down by adding 50 μl of Protein-A/G-Sepharose beads and incubated for 6 h at 4˚C. The beads were washed for 10 min each with the lysis buffer followed by high-salt wash buffer (0.1% SDS, 1% Triton X-100, 2 mM EDTA, 20 mM Tris-HCl [pH 8.1], 500 mM NaCl), LiCl wash buffer (250 mM LiCl, 1% NP-40, 1% deoxycholate, 1 mM EDTA, 10 mM Tris-HCl [pH 8.0]), and TE buffer (10 mM Tris-HCl [pH 8.0], 1 mM EDTA). Finally, DNA was eluted with elution buffer (1% SDS, 100 mM NaHCO$_3$). To reverse the formaldehyde cross-linking, elutes were incubated at 65˚C overnight with the addition of 5 M NaCl to a final concentration of 200 mM and later digested at 56˚C for 4 h with Proteinase K at a final concentration of 50 μg/ml. Protein extraction was carried out by phenol/chloroform extraction and DNA was precipitated by ethanol. The resulting sheared DNA fragments were used as templates in semi-quantitative PCR analysis. Purified DNA samples were quantified, and PCR was performed with 20 ng of DNA. The sequences of the PCR primers are mentioned in S6 Table. Quantification of all PCR products was done by using ImageJ software.

## Subcellular fractionation

After treatment of U2OS cells with increasing dose of Etoposide, the cytosolic protein fraction was removed by incubation in hypotonic buffer (10 mM HEPES, pH 7, 50 mM NaCl, 0.3 M sucrose, 0.5% Triton X-100, supplemented with protease inhibitor; Roche) for 15 min on ice and centrifuged at 1500xg for 5 min. The soluble nuclear fraction was removed by incubation with nuclear buffer (10 mM HEPES, pH 7, 200 mM NaCl, 1 mM EDTA, 0.5% NP-40 and protease inhibitor cocktail) for 10 min on ice and then centrifuged at 16000xg for 2 min. The pellets were resuspended in lysis buffer (10 mM HEPES, pH 7, 500 mM NaCl, 1 mM EDTA, 1% NP-40 and protease inhibitor cocktail), sonicated at low amplitude and centrifuged for 1 min at 16000xg; the supernatant was then transferred to a new tube. Whole-cell lysates for loading control were prepared by lysis in RIPA buffer (150 mM NaCl, 50 mM Tris-HCl [pH 8], 1% NP-40, 0.1% SDS, 0.5% sodium deoxycholate, 5% glycerol) supplemented with protease inhibitor (Roche).

## Immunoblotting and antibodies

Standard Bradford assay was used to estimate protein concentrations. Proteins were resolved on a 7.5% SDS-PAGE gel and transferred onto PVDF membranes (Millipore) using Trans-Blot SD Semi-Dry Transfer Cell (Bio-Rad). The membranes were blocked using 3% Bovine serum albumin (BSA) (w/v) in TBST (50 mM Tris-HCl pH 8.0, 150 mM NaCl, 0.1% Tween-20). The membranes were then incubated with appropriate primary antibodies (S6 Table) overnight at 4˚C. The membranes were washed with TBST and incubated with respective HRP-conjugated secondary antibodies (1:8000; Santa Cruz) for 1 h at 4˚C. After TBST washes, membranes were developed with chemiluminescent HRP substrate (Millipore) and imaged using GE healthcare LAS 4000 Chemidoc.

## Cell survival assays

Cell survival assays were performed as described previously [76]. Briefly, cells were mock-treated or treated with 4-OHT for 4 h, followed by recovery in fresh medium. Cell survival was

monitored 4–5 days post-recovery by MTT assay using a microplate reader (VersaMax ROM version 3.13). Percentage cell survival was calculated as treated cells/untreated cells × 100.

## Quantification and Statistical Analysis

Statistical differences in immunostaining experiments, qPCR assays for measurement of ssDNA generated by end resection, ChIP analysis, cell survival, HR and NHEJ assays were determined in terms of p-value from two-tailed *Student's* t-test of unequal variance. The numerical data for all the graphs in this study is available in S1 Data.

## Supporting information

**S1 Fig. Methodology for quantitative measurement of end resection, cell cycle synchronization and knockdown analyses of various proteins.** (A) Design of qPCR primers for measurement of DSB% at two *Asi*SI sites (red arrows: DSB1 and DSB2) located on Chromosome 1 and measurement of resection at sites adjacent to the *Asi*SI sites (black arrows) [24]. The primers on Chromosome 22 ('No DSB') were used as negative control. The primer pairs for 'DSB1' are across *Ava*I and *Bsr*GI restriction sites; and for 'DSB2' are across *Nme*AIII and *Bam*HI restriction sites. The primer pair for 'No DSB' is across a *Hin*dIII restriction site. (B) Experimental design for cell synchronisation by RO-3306 at S/G2 phase for measurement of end resection (detailed protocol in Materials and Methods section). (C) Validation of shRNA mediated knockdown of indicated proteins by immunoblotting with respective antibodies after 48 h and MCM3 as loading control.
(TIF)

**S2 Fig. FANCJ promotes DNA end resection.** (A) ER-*Asi*SI U2OS cells depleted for the indicated proteins were treated with zeocin (1μg/ml) for 4 h or mock treated. Cells were fixed and stained with γ-H2AX and pRPA2 (S4/S8) antibodies to detect ssDNA generated by end resection. Representative image for γ-H2AX and pRPA2 (S4/S8) foci are shown. (B) Graph represents the mean fluorescence intensity of γ-H2AX and pRPA2 (S4/S8) foci/nucleus from indicated cells in (A). N = 3; error bars indicate standard deviation (SD) and statistical significance was measured by two-tailed *Student's* t-test of unequal variance. *p < 0.05; **p < 0.01; ***p < 0.001; N.S., non-significant. (C) ER-*Asi*SI U2OS cells treated with either control shRNA, shFANCJ #1 or shCtIP were treated with increasing dose of zeocin (0, 0.5 and 1 μg/ml) for 4 h. Whole cell lysates were separated on 10% SDS-PAGE and probed for the indicated proteins to measure their damage induced enrichment in the cell. (D) ER-*Asi*SI U2OS cells depleted for the indicated proteins were treated with zeocin (1μg/ml) for 4 h or mock treated. Cells were fixed and stained with γ-H2AX and RAD51 antibodies. Representative image for γ-H2AX and RAD51 foci are shown. (E) Graph represents the mean fluorescence intensity of γ-H2AX and RAD51 foci/nucleus from indicated cells in (D). N = 3; error bars indicate standard deviation (SD) and statistical significance was measured by two-tailed *Student's* t-test of unequal variance. *p < 0.05; **p < 0.01; ***p < 0.001; N.S., non-significant. (F) FANCJ depleted ER-*Asi*SI U2OS cells were treated with 300 nM 4-OHT for 2 h or mock treated, and ChIP assays were performed using antibody directed against RAD51. ChIP efficiencies (as percent of input immunoprecipitated) were measured by semiquantitative PCR at 80 bp from *Asi*SI induced DSB1 site. N = 3, with error bars indicating SD and statistical significance was measured by two-tailed *Student's* t-test of unequal variance. *p < 0.05; **p < 0.01; ***p < 0.001; N.S., non-significant. (G) ER-*Asi*SI U2OS cells depleted for the indicated proteins were treated with zeocin (1μg/ml) for 4 h or mock treated. Cells were fixed and stained with γ-H2AX and RPA70 antibodies. Representative image for γ-H2AX and RPA70 foci are shown.

(H) Graph represents the mean fluorescence intensity of γ-H2AX and RPA70 foci/nucleus from indicated cells in (G). N = 3; error bars indicate standard deviation (SD) and statistical significance was measured by two-tailed *Student*'s t-test of unequal variance. $^{*}p < 0.05$; $^{**}p < 0.01$; $^{***}p < 0.001$; N.S., non-significant. (I) U2OS-SCR 18 cells treated with either control shRNA or shFANCJ #1 were treated with increasing dose of Etoposide (0,1 and 4μM), followed by subcellular fractionation. Chromatin enriched fractions were separated on 10% SDS-PAGE and probed for the indicated proteins to measure damage induced recruitment to chromatin. Whole cell lysates indicate total protein levels. (J) ER-*Asi*SI U2OS cells depleted for the indicated proteins were treated with zeocin (1μg/ml) for 4 h or mock treated. Cells were fixed and stained with γ-H2AX and FANCJ antibodies. Representative image for γ-H2AX and FANCJ foci are shown. (K) Graph represents the mean fluorescence intensity of γ-H2AX and FANCJ foci/nucleus from indicated cells in (J). N = 3; error bars indicate standard deviation (SD) and statistical significance was measured by two-tailed *Student*'s t-test of unequal variance. $^{*}p < 0.05$; $^{**}p < 0.01$; $^{***}p < 0.001$; N.S., non-significant.
(TIF)

**S3 Fig. Relative abundance of exogenous vs endogenous FANCJ.** (A)Relative protein levels of endogenous FANCJ and WT/S990A/S990E-HA-6xHis-FANCJ. (B) Relative protein levels of endogenous FANCJ and WT/K1249R/K1249Q-HA-6xHis-FANCJ. (C) Relative protein levels of endogenous FANCJ and WT, K52A, K52R, K52A/1249R- HA-6xHis-FANCJ. In (A), (B) and (C), western blotting was carried out using FANCJ specific antibody.
(TIF)

**S1 Table. Sequences of qPCR Primers used for studying ssDNA generation.**
(PDF)

**S2 Table. DSB% at DSB1 and DSB2 sites.**
(PDF)

**S3 Table. Sequences of primers used in this study.**
(PDF)

**S4 Table. Sequence of shRNAs used in this study.**
(PDF)

**S5 Table. List of Antibodies used in this study.**
(PDF)

**S6 Table. Sequences of Primers used in this study for ChIP-PCR.**
(PDF)

**S1 Data. Numerical data for all the graphs in this study.**
(XLSX)

## Acknowledgments

We thank Drs. Gaelle Legube and Sharon Cantor for generously providing ER-*Asi*SI system containing U2OS cells and FANCJ cDNA plasmid, respectively. We thank Drs. Sagar Sengupta and Kumar Somyajit for their useful comments. We thank Sneha Saxena, Kaustubh Shukla, Suruchi Dixit and Shivakumar Basavaraju for their critical reading of the manuscript. We thank Ranjitha and Lipika from the divisional FACS facility for their help. We also thank Puja Biswas for her technical help with microscopy experiments.

## Author Contributions

**Conceptualization:** Sarmi Nath, Ganesh Nagaraju.

**Data curation:** Sarmi Nath, Ganesh Nagaraju.

**Formal analysis:** Sarmi Nath.

**Funding acquisition:** Ganesh Nagaraju.

**Investigation:** Sarmi Nath.

**Methodology:** Sarmi Nath.

**Project administration:** Ganesh Nagaraju.

**Resources:** Ganesh Nagaraju.

**Software:** Sarmi Nath.

**Supervision:** Ganesh Nagaraju.

**Validation:** Sarmi Nath.

**Visualization:** Sarmi Nath.

**Writing – original draft:** Sarmi Nath, Ganesh Nagaraju.

**Writing – review & editing:** Sarmi Nath, Ganesh Nagaraju.

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
