## [Decision Letter · Decision Letter 0]

3 Dec 2019

Dear Dr NAGARAJU,

Thank you very much for submitting your Research Article entitled 'FANCJ helicase promotes DNA end resection by facilitating CtIP recruitment to the DNA double-strand breaks' to PLOS Genetics. Your manuscript was fully evaluated at the editorial level and by independent peer reviewers. The reviewers appreciated the attention to an important problem, but raised some substantial concerns about the current manuscript. Based on the reviews, we will not be able to accept this version of the manuscript, but we would be willing to review again a much-revised version. We cannot, of course, promise publication at that time.

If you decide to revise the manuscript for further consideration at PLOS Genetics, please aim to resubmit within the next 60 days, unless it will take extra time to address the concerns of the reviewers, in which case we would appreciate an expected resubmission date by email to plosgenetics@plos.org.

[LINK]

We are sorry that we cannot be more positive about your manuscript at this stage. Please do not hesitate to contact us if you have any concerns or questions.

Yours sincerely,

Wolf-Dietrich Heyer

Guest Editor

PLOS Genetics

Gregory P. Copenhaver

Editor-in-Chief

PLOS Genetics

Dear Ganesh

Thank you for submitting your work to PLoS Genetics. Your manuscript was reviewed by three experts in the field and their individual comments are attached below. Moreover, I also read your manuscript with great interest and have a few specific comments, of which some relate to comments made by the other reviewers. All reviewers consider your work of potential importance and interest for this journal, but they suggest major additions and revisions to better the support the conclusions and mechanistic interpretations. I agree with their general assessment and hope that you can revise your manuscript accordingly.

The reviewers make a partially overlapping set of suggestions, and I do not think that every experimental suggestion has to be implemented in the revision. In general, I think it is important to make sure that the major conclusions are firm and corroborated by the data.

Here are my suggestions for the revised version of your manuscript:

Reviewer 1: This reviewer makes a number of excellent suggestions that would strengthen the mechanistic interpretation. While many points are listed, in my mind the major points you should focus on are:

1) Further analysis of the point mutants. See Minor Points #4, 5, 6, 7.

2) Additional Western blot controls (CHK2, gH2AX, RPA S4/8).

3) Survival assays would add significantly and appear possible in the AsiSI system.

RPA and RAD51 focus analysis would complement the ChIP and fractionation analysis but is not essential. I do not think that another DSB inducing agent needs to be used, as you already show that etoposide shows the expected decrease in chromatin association of RPA70 and RAD51.

About the Minor points listed: 1) Please include more discussion, but no new experiments are needed. 2) If IF is possible, this could strengthen the current interpretation. 3) This is not essential for the revision. 4-6) These are the major experimental additions. 7) Please focus on the FANCJ mutants.

Of the ‘Additional’, point a is a good suggestion but not needed for this revision. The remainder should pose no problem to implement.

Reviewer 2: The reviewer makes several specific experimental suggestions: 1) FANCJ-K1249Q-CtIP interaction. 2) Differentiation of two interaction models. Results from both experiments would add to the present data and strengthen the mechanistic model.

The comment about statistical significance must be addressed. The minor comments should not pose a problem.

Reviewer 3: The reviewer makes an important point whether FANCJ is directly involved in resection or via recruitment of CtIP. I agree that the phenotype of the Walker A box mutants cannot fully settle this question. The analysis of the FANCJ-K52A mutant in the CtIP recruitment assay is an excellent suggestion. Also, the NHEJ analysis and HR analysis with the FANCJ-K1249R mutant. I think the suggested epistasis analysis with BRCA1 and 53BP1 is not needed for this manuscript. The last 4 points should not pose a problem to address with additional explanation.

Editor comments:

1) In the discussion, I think you have to relate your observations on resection back to your previous finding that FANCJ affects the balance between long and short tract conversion.

2) Please consult this paper (Juhasz et al. 2018 Molecular Cell) on HR in U2OS cells and consider the impact on the present and previous data.

3) I agree with reviewer 2, that the results with FANCJ KR and KA mutants needs to be presented and discussed more carefully. Has it been experimentally shown that KR binds but not hydrolyzes ATP and that KA does not bind ATP? In fact, some KA mutants bind ATP but do not lead to a conformational change that is usually accompanied by ATP binding. Moreover, in my mind the data show a differential requirement for both, KA and KR, suggesting that motor activity is needed but also conformational change after ATP binding (given the caveats above). This may suggest a 2-step model. This issue (also Rev. 2 comment on this topic) needs rewording and more discussion, but not new experiments.

I hope these comments will be helpful for the revision and look forward to the new version of the manuscript.

My best regards

Wolf

Reviewer's Responses to Questions

**Comments to the Authors:**

Reviewer #1: The article titled’ FANCJ helicase promotes DNA end resection by facilitating CtIP recruitment to the DNA double-strand breaks’ provides insight into the role of FANCJ in promoting initial nucleolytic processing of DSBs through FANCJ interaction with BRCA1 and CtIP, components of the homologous recombination pathway of double-strand break repair. The authors claim that FANCJ has a direct role in the recruitment of CtIP through acetylation by CBP after localisation to DSBs in a BRCA1 dependent manner through phosphorylation by CDK. Furthermore, the authors claim that the helicase activity of FANCJ is also required for DNA resection. These claims are generally not very well supported and require significant amount of experimental work to solidify the hypothesis.

Major comments:

The authors provide insight into the impact of FANCJ on ssDNA generation through resection primarily using the ER-AsiSi system. While the author’s provide an initial additional read-out for ssDNA through the analysis of BRDU foci (unusually high in control given the number of DSBs generated by tamoxifen-dependent induction of ER-AsiSi), it is essential that the authors conduct similar analysis to complement the findings associated with separation of function point mutants identified. In addition, (i) the inclusion of additional controls in AsiSi DSB resection efficiency analysis is essential such as (ii) western blots showing level of RPA S4/8, CHK2 and gH2AX for FANCJ-deficient, CtiP-deficient (internal control) and the doubles. These should be done in response to Tamoxifen treatment as well as to a DSB inducing agent, such as zeocin. It is somehow surprising the level of foci shown in Figure 1B and supplementary figure 2a upon tamoxifen treatment (BrDU and pRPA2 S4/S8, gH2AX) given that such treatment should generate relatively small number of DSBs – do the authors see the same pattern of response upon zeocin treatment?) – this would be an essential analysis to include (also for the separation of function mutants). Since defect in resection would result in a defective RPA and RAD51 foci formation these should be analysed as well (upon tamoxifen as well as zeocin treatment).

Finally, survival assays for FANCJ and the separation of function mutants should be included in response to tamoxifen as well as zeocin (or another DSB inducing agent).

Minor points to address:

1. Figure 2A shows that shMRE11 results in a significant defect in FANCJ recruitment to DSBs which may support previous reports. However, an explanation for this is not addressed throughout the article. Why does MRE11 deficiency affect FANCJ recruitment- what is the mechanism and evidence for this in the authors hands or from the literature? This should be included in text.

2. Immunofluorescence microscopy for FANCJ recruitment to AsiSi breaks should complement ChIP and resection analysis in Figure 2 (see also major comments above).

3. Figure 2B and C: BARD1/ BRCA1-FANCJ interaction appears to be upregulated after DSB induction. Given that they form a stable complex and together are shown to counteract 53bp1 function to promote DNA resection (Densham et al., 2016. Nat. Struc. Biol), including BARD1 and BRCA1 in ChIP analysis (Figure 2C) to determine whether FANCJ depletion affects their recruitment is advised.

4. Resection analysis in Figure 4C and 5C should include the C�-FANCJ mutant as comparison to separate function to BRCA1 and CtIP, this will enable insight into relative contribution to DNA resection based on BRCA1/CtIP- FANCJ nteraction vs interaction with additional DNA resection factors (specifically MRE11). This will underline the role of the identified point mutations and FANCJ in the wider mechanism of HR.

5. Additional experimental read-outs for resection defects are required for all separation of function mutants analysed: FANCJ S990A/E, K1249R/Q, CtIP-T847A, FANCJ K52A/R. BRDU foci analysis is proposed to support resection efficiency assays.

6. Figure 7 C displays resection efficiency after depletion of CtIP followed by rescue analysis using mutants. Resection analysis should include shFANCJ and shFANCJ-K1249R to show epistasis or relative contribution of complexes to resection efficiency.

7. Figure 8 C: Resection analysis should include FANCJ-S990A and K1249R separation of function mutants or CtIP or BRCA1 shRNA treatments. If the interactions with CtIP or BRCA1 is epistasic to the helicase activity. Authors may also include K52A + K1249R FANCJ double point mutants as additional control to demonstrate epistasis. This will solidify the mechanism described.

Additional:

a. Use of CDK or CBP inhibitors followed by analysis of interactions with CtIP will reinforce the claim of their direct role in regulating FANCJ.

b. Figure S3C should be included in Figure 2 as additional panel.

c. Figure S4A should be included in Figure 3 as additional panel.

Thorough checking of sentence structure and grammar is required throughout article.

Reviewer #2: In this study the authors report on the interaction of FANCJ with CtIP that they found to be required for the recruitment of CtIP to DNA double-strand breaks (DSBs). They further show that the interaction with CtIP is dependent on FANCJ acetylation at K1249, which in turn requires S990 phosphorylation.

Overall, the experiments in this paper are carefully conducted and the results are clearly presented.

Major comments:

– The authors show that the interaction between FANCJ and CtIP depends on DSBs (Figure 2B), but do not show whether this is due to acetylation of FANCJ in response to DSBs. If so, FANCJ K1249Q should be able to interact with CtIP also in the absence of DSBs. The authors should test this.

– The authors show that phosphorylation of FANCJ at S990 is required for acetylation at K1249 and that both FANCJ S990A and K1249R are unable to interact with CtIP. In my view there are two possible interaction modes between FANCJ and CtIP. First, phosphorylation of S990 is required for K1249 acetylation (for example through recruitment of CBP or another acetyltransferase), but CtIP physically only binds to acetylated K1249; or second, phosphorylation of S990 is required for K1249 acetylation and CtIP binds to both phosphorylated S990 and acetylated K1249. I would encourage the authors to utilise a double mutant (S990A/K1249Q) and do CtIP interaction experiments (similar as in Figure 6A) to distinguish between these two possibilities.

– It is interesting that the authors see different phenotypes with FANCJ K52A and K52R. However, they should be more careful with their wording (results section and abstract). Given that they see a partial rescue with K52R, it would appear that ATP binding (as opposed to ATP hydrolysis and helicase activity) is sufficient to partially restore resection.

– Figure 8C and S7A: Statistical significance should be calculated between “shFANCJ + WT” and “shFANCJ + K52A” or “shFANCJ + K52R” and not between “shFANCJ + K52A” and “shFANCJ + K52R”.

Minor comments:

– Introduction, line 90: It should read “maintenance of microsatellite stability”

– Figure 1D: The authors should mention somewhere why they include knock-down of DNA2 and 53BP1.

– Results, line 198: It should read “FANCJ K1249R was defective…” not “ FANCJ K1249R expressing cells…”

– Results, lane 199-201: Since the authors show before that CtIP is not required for recruitment of FANCJ to DSBs, it is not so surprising that FANCJ K1249R can still localise to DSBs. Please rephrase.

– Table S4: Change title (since it contains not only SDM primers) and highlight all nucleotide changes for SDM primers.

Reviewer #3: Previous studies have shown a decreased frequency of homologous recombination (HR) in the absence of FANCJ. In this study, the authors investigate the role of FANCJ in the earliest step of HR, DNA end resection. They find reduced generation of single-stranded DNA (ssDNA) after induction of AsiSI endonuclease and show lower accumulation of CtIP at DSBs in FANCJ-depleted cells. Furthermore, they find that CtIP recruitment/retention at DSBs requires phosphorylation and acetylation of FANCJ, and provide evidence that the CtIP-FANCJ interaction is independent of BRCA1, a known interaction partner of both CtIP and FANCJ.

The findings presented support a role for FANCJ in recruitment of CtIP to DSBs to facilitate end resection. It is unclear from the data whether FANCJ plays a direct role in end resection or if its primary function is in recruitment of CtIP. The end resection defect of the FANCJ-K52A mutant would appear to suggest a catalytic role, but it is not obvious why MRN-CtIP would require a helicase to initiate end resection. Is CtIP recruited to DSBs in cells expressing FANCJ-K52A? If FANCJ is acting at an early step it would be expected to shift the balance from HR to NHEJ. I suggest the authors measure NHEJ efficiency in FANCJ depleted cells. I also recommend measuring HR in cells expressing the FANCJ-K1249R mutant. To confirm that FANCJ functions independently of BRCA1 in end resection, I suggest measuring end resection in sh53BP1 shFANCJ cells and comparing to sh53BP1 shBRCA1 cells.

Most of the assays rely on induction of DSBs by AsiSI. Is there equivalent efficiency of AsiSI cleavage at DSB 1 and 2 in the all of the cell lines expressing shRNAs? Were the values of ssDNA shown for the end resection assay normalized to the % AsiSI cleavage at each site?

Most of the resection assays show a single time point, 4 h, after DSB induction. Were other time points analyzed? Does resection in FANCJ-depleted cells increase at later time points? Why were ChIP assays done 2 h after 4-OHT treatment? Later time points should also be used to measure RAD51 association with DSBs.

The methods for transfection with shRNA constructs are not explained in enough detail. How long after transfection were cells treated with 4-OHT to induce AsiSI? Do cells remain arrested in G2 after transfection with shRNAs and induction of AsiSI? At what time point were Western blots performed to assess knockdown efficiency?

Line 150: I don’t think the authors can conclude direct interaction based on IP results. The authors would need to demonstrate interaction between the purified proteins to conclude that it is direct.

**Have all data underlying the figures and results presented in the manuscript been provided?**

Reviewer #1: Yes

Reviewer #2: Yes

Reviewer #3: Yes

PLOS authors have the option to publish the peer review history of their article (what does this mean?). If published, this will include your full peer review and any attached files.

Reviewer #1: No

Reviewer #2: No

Reviewer #3: No

---

## [Decision Letter · Decision Letter 1]

17 Feb 2020

Dear Ganesh

All reviewers appreciate the effort that was invested in the revision and agree that their concerns were addressed. In principle, I am happy to accept the revision but reviewers 1 and 2 note some minor issues that can be addressed with text changes and editing. Please make the changes accordingly and submit a second revision along with a short description, how you implemented the reviewers points. Congratulations on a nice piece of work.

[GPC note - one or more of the reviewers noted numerous grammatical errors in the text of the manuscript. PLOS Genetics does not offer copy editing services, so we ask that you carefully proof the manuscript at this stage.  Some authors find it helpful to have an experienced colleague edit for grammar (not science), while other find professional editing services helpful.]

Below, please find a standard "minor revision" decision letter.

---

Thank you very much for submitting your Research Article entitled 'FANCJ helicase promotes DNA end resection by facilitating CtIP recruitment to the DNA double-strand breaks' to PLOS Genetics. Your manuscript was fully evaluated at the editorial level and by independent peer reviewers. The reviewers appreciated the attention to an important topic but identified some aspects of the manuscript that should be improved.

We therefore ask you to modify the manuscript according to the review recommendations before we can consider your manuscript for acceptance. Your revisions should address the specific points made by each reviewer.

[LINK]

Yours sincerely,

Wolf-Dietrich Heyer

Guest Editor

PLOS Genetics

Gregory P. Copenhaver

Editor-in-Chief

PLOS Genetics

Reviewer's Responses to Questions

**Comments to the Authors:**

Reviewer #1: The manuscript has been now substantially improved and I would like to congratulate the authors on this very nice pice of work.

Minor comments:

1. Canonical image for Fig S2A shCtIp pRPAS4/S8 stain does not support graphical representation in Fig S2B.

2. shCtIP canonical image for RAD51 staining (Fig S4C) does not complement graph trend (Fig S4D).

Reviewer #2: The authors did a very good job in addressing the reviewers' comments with many new experiments. I think they report important findings that will be of great interest to the readership of PLOS Genetics.

However, I still have a few remaining criticisms:

– I still think that the issue KA/KR is not properly addressed. For example, in the abstract, the authors write "its helicase activity is crucial for promoting end resection”. Given Figure 9D where the authors see a partial restoration of ssDNA with the KR (but not the KA) variant, this is not correct. They should instead use the term “intact ATP binding domain” or “ATP binding”. Same problem in the results and discussion sections.

– I think the manuscript requires careful copy-editing. There are issues with grammar, hyphenation, articles etc.

– I also think that the flow of the paper could be improved – ironically, I think this problem comes partially from the wealth of new data that were added. It would certainly help to add more sub-headings to the results section and to try and integrate the SI data in a better way.

Minor:

– Not all SI figures come in the right order in the text.

– Figure S6A/B: I would swap red/green in the graph to be consistent.

– Figures S12: Please align MRE11 blot.

Reviewer #3: The authors have adequately addressed all of my concerns.

**Have all data underlying the figures and results presented in the manuscript been provided?**

Reviewer #1: Yes

Reviewer #2: Yes

Reviewer #3: Yes

PLOS authors have the option to publish the peer review history of their article (what does this mean?). If published, this will include your full peer review and any attached files.

Reviewer #1: No

Reviewer #2: No

Reviewer #3: No

---

## [Editor Report · Decision Letter 2]

28 Feb 2020

Dear Ganesh,

We are pleased to inform you that your manuscript entitled "FANCJ helicase promotes DNA end resection by facilitating CtIP recruitment to DNA double-strand breaks" has been editorially accepted for publication in PLOS Genetics. Congratulations!

Yours sincerely,

Wolf-Dietrich Heyer

Guest Editor

PLOS Genetics

Gregory P. Copenhaver

Editor-in-Chief

PLOS Genetics

Comments from the reviewers (if applicable):

**Data Deposition**

http://datadryad.org/submit?journalID=pgenetics&manu=PGENETICS-D-19-01831R2

**Press Queries**

---

## [Editor Report · Acceptance letter]

27 Mar 2020

PGENETICS-D-19-01831R2 

FANCJ helicase promotes DNA end resection by facilitating CtIP recruitment to DNA double-strand breaks 

Dear Dr NAGARAJU, 

We are pleased to inform you that your manuscript entitled "FANCJ helicase promotes DNA end resection by facilitating CtIP recruitment to DNA double-strand breaks" has been formally accepted for publication in PLOS Genetics! Your manuscript is now with our production department and you will be notified of the publication date in due course.

With kind regards,

Kaitlin Butler

PLOS Genetics

On behalf of:
